# Geometry & Optimization of Three-Layer Networks: Symmetry Breaking as a Unifying Principle

## Abstract

We propose symmetry breaking as a unifying principle underlying geometric and optimization phenomena in the training of fully connected three-layer networks. First, we demonstrate the prevalence of critical points that break symmetries jointly induced by the loss, network architecture, and data distribution, in direct agreement with theoretical predictions. Group-theoretic results, seemingly far removed, are then shown to govern the structure of the Hessian and Gauss–Newton matrices, with empirical phenomena characteristic of deep learning—such as the bulk-and-outliers spectrum and optimization trajectories concentrating in low-dimensional subspaces—emerging naturally as manifestations of symmetry breaking. Leveraging the rich symmetry structure, we employ group representation-theoretic techniques to derive sharp estimates of the eigenspectrum in high dimensions, requiring only a small, fixed subset of Hessian entries. The analysis reveals notable curvature differences between local and global minima, contrary to the analogous two-layer setting, which point to a possible dependence of the flat minima conjecture on network depth.

## 1 Introduction

Optimization problems in deep learning (DL) entail multiple sources of difficulty: they are nonconvex, high-dimensional, and stochastic. Yet, neural networks are routinely trained to solve complex learning problems. Empirical studies seeking to explain why training is possible, despite its apparent intractability from the standpoint of classical optimization theory, have uncovered striking phenomena related to curvature, optimization, and generalization that arise consistently across diverse architectures and datasets. Building on recent theoretical developments Arjevani (2024), we propose symmetry breaking (SB) as a framework for elucidating and unifying several of these key phenomena in three-layer networks.

A natural starting point for exploring loss landscapes is the Hessian matrix, as it captures essential aspects of local curvature. Extensive empirical studies consistently report a distinctive spectral structure emerging early in training: a bulk of eigenvalues clustered near zero, accompanied by a small number of large outliers LeCun et al. (2002); Sagun et al. (2016; 2017); Ghorbani et al. (2019); Papyan (2018), henceforth referred to as the *bulk–outlier* structure. This phenomenon is not limited to the Hessian: similar bulk–outlier structure has also been observed for the Gauss–Newton (GN) matrix Zhang et al. (2019); Jastrzebski et al. (2020), and appears even when spectra are examined layer by layer Li et al. (2020a); Papyan (2020). Empirical work Gur-Ari et al. (2018); Ghorbani et al. (2019) has further demonstrated that, in conjunction with the formation of the bulk–outlier structure, gradients tend to concentrate in the low-dimensional subspaces spanned by the Hessian's outlier eigenspaces early in training. The identification of the bulk–outlier phenomenon was significant both for highlighting factors that strongly influence optimization, further motivating curvature-aware algorithms such as Foret et al. (2020); Chaudhari et al. (2019) and others discussed below, and for suggesting latent structure shaping the loss landscape in DL. Despite substantial work on possible mechanisms that may account for the bulk–outlier phenomenon, the problem appears to remain open, as we briefly review below.

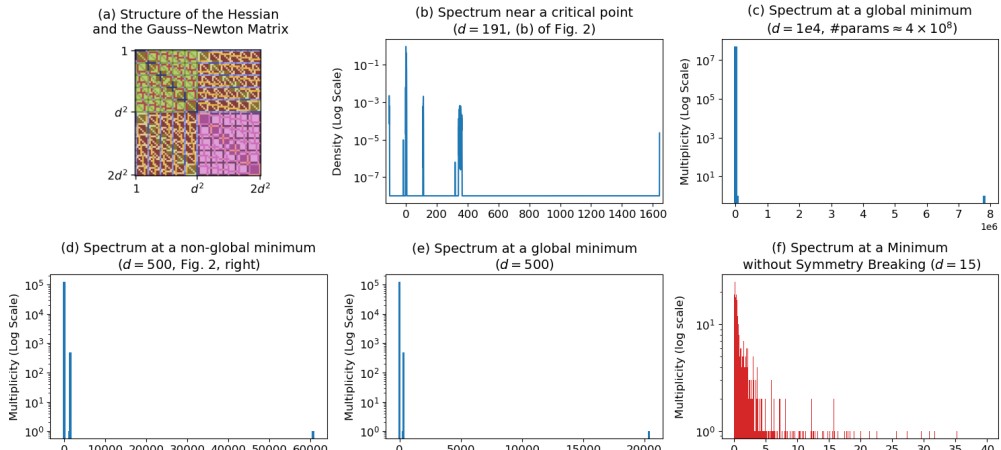

Figure 1: Symmetry breaking (SB), relative to symmetries *jointly* induced by the loss, architecture and data, shapes the structure and eigenspectra of the Hessian and Gauss–Newton matrices.
(a) At SB points of the loss (2.3), both matrices lie in the same fixed low-dimensional space, here of dimension 37. Their spectra (b–e), shown for the Hessian, are in direct agreement with the structure dictated by Theorem 1: $\Theta(1)$ eigenvalues with multiplicity $\Theta(d^2)$, $O(d)$ eigenvalues with multiplicity $\Theta(d)$, and the remaining eigenvalues each with multiplicity $\Theta(1)$. Without symmetry breaking (f), the framework remains applicable but, correctly, does not predict the *bulk–outlier* structure; here, samples were labeled by a target network with random weights. Thus, in our setting, quantitative theoretical predictions based on SB provide a consistent account of the spectral form empirically observed across a wide range of network architectures and datasets (both at convergence and early in training; see Figure 3). (c) Approximate spectrum near a critical point (Lanczos Meurant & Strakovs (2006), PyHessian Yao et al. (2020)). (d) SB further enables precise characterization of the spectrum at SB points, even in extremely high-dimensional settings. (d, e) The maximum eigenvalues at non-global minima far exceed those at the global minimum, supporting the flat minima conjecture.

One general approach offering mechanisms for understanding spectral behavior is random matrix theory (RMT) Anderson et al. (2010); Tao (2012). In this framework, the Hessian or Gauss–Newton matrices, whether considered in general or derived under specific model assumptions, are treated as *random draws* from a distribution, for instance from the canonical Wigner or Wishart ensembles Choromanska et al. (2015a); Baskerville et al. (2021; 2022b); Granziol et al. (2022); Pennington & Worah (2017); Pennington & Bahri (2017). In many cases, RMT is then capable of providing detailed characterizations of the average spectral density in the infinite-dimensional limit. However, the resulting theoretical insights fall short of capturing the empirically observed bulk–outlier structure, with major discrepancies reported Granziol (2020); Baskerville et al. (2022a). The modeling assumptions, motivated by studies of *glassy* systems, have also been criticized for their predicted dynamics Baity-Jesi et al. (2018). Developing more suitable random matrix models (see Baskerville et al. (2022c) for recent progress) or exploring alternative modeling assumptions Choromanska et al. (2015b) remains an active research direction.

Another large body of work aimed at explaining the bulk–outlier spectra has focused on the GN matrix, as a means to capture additional important aspects of curvature. The GN matrix is often introduced as an approximation to the Hessian in nonlinear least squares problems, obtained by discarding terms involving second-order derivatives, but keeping first-order derivatives, of the model output with respect to the parameters. For the squared loss and network models $N(\cdot; W)$ parameterized by weights $W$ ranging over the weight space $\mathbb{W}$, the GN matrix takes the form $H_{\mathrm{GN}}(W) = 2\mathbb{E}_{\mathbf{x}\sim\mathcal{D}} \left[ \nabla_W N(\mathbf{x}; W)\nabla_W N(\mathbf{x}; W)^\top \right]$, where $\mathcal{D}$ denotes the (empirical or true) input distribution. In the study of the bulk–outlier phenomenon, the GN matrix has been employed either through the exploitation of the low-rank structure of the empirical GN Sagun et al. (2017) or through interpretations that seek to attribute certain outlying eigenvalues to those of the GN matrix Papyan (2018; 2019; 2020); Cohen et al. (2021). While these observations offer refinements to the bulk–outlier phenomenon, they do not yet provide mechanisms or causal accounts (see Figure 1).

The GN matrix also plays an important role in continuous optimization, being computationally cheaper than the Hessian, always positive semidefinite (and thus more stable as a preconditioner), and approaching the Hessian as the loss decreases. For these reasons, it has been commonly used in optimization methods that aim to exploit curvature in a computationally efficient way Dennis Jr & Schnabel (1996); Nocedal & Wright (2006); Tran-Dinh et al. (2020), as well as in convergence analyses Wu & Su (2023). In deep learning, the computational burden of working directly with the high-dimensional Hessian is particularly prohibitive, motivating the use of more tractable approximations LeCun et al. (2002); Martens et al. (2010); Martens & Sutskever (2012). Among these, and for the reasons indicated above (see also (Martens, 2020, Section 8.1)), the GN matrix has emerged as a prominent alternative Park et al. (2000); Desjardins et al. (2015); Martens & Sutskever (2011); Botev et al. (2017); Martens & Grosse (2015); Gargiani et al. (2020); Korbit et al. (2024); Vinyals & Povey (2012). Under certain metrics, the GN is not merely an approximation but rather serves as the exact analogue of the Hessian. This is the case with the natural gradient descent algorithm Amari (1998), where the Fisher information (FI) matrix Rao (1992), used as a preconditioner Heskes (2000), coincides in many important cases with the GN (and its generalized version Schraudolph (2002)) Martens (2020); Pascanu & Bengio (2013); Heskes (2000). Yet another important role of the GN matrix arises from its definition as the (non-centered) covariance matrix of the gradients Xing et al. (2018); Gupta et al. (2018); Liu et al. (2023), a perspective that naturally links it to dynamics and generalization, e.g., MacDonald et al. (2023). All of the above is intended to illustrate the significant role of the GN matrix and to underscore the importance of understanding its structure and spectral properties, see Figure 1 for a demonstration of our results in the present setting.

Similar to the bulk–outlier phenomenon, a range of findings on dynamics and curvature indicate certain features of the eigenspectra that may stabilize early in training, particularly with respect to step size and batch size Jastrzebski et al. (2020); Jastrzkebski et al. (2018); Cohen et al. (2021). Curvature, beyond its evident connection to dynamics, has long been debated in relation to generalization, with flat minima, under certain notions of flatness, being associated with improved performance Hochreiter & Schmidhuber (1997); Keskar et al. (2016); Jastrzkebski et al. (2017); Simsekli et al. (2019); Chaudhari et al. (2019). Thus, although the loss landscape contains minima that do not generalize well Zhang et al. (2016), in practice optimization algorithms appear to often avoid them. A common explanation is that these algorithms are biased toward flat minima, which, owing to their statistical stability, tend to generalize better. The precise notion of flatness has been shown to present several difficulties Dinh et al. (2017), and despite progress in understanding it as well as other forms of *implicit bias* Neyshabur et al. (2014); Gunasekar et al. (2018), the mechanisms underlying generalization in DL remain only partially understood. In this work, we demonstrate at a high conceptual level that regarding SB as a form of implicit bias can yield tighter and practically relevant generalization bounds.

Taken together, these strands of phenomena highlight both progress and persistent gaps in the theory of DL, which we propose to study and interpret through SB. Our exposition centers on fully connected three-layer models, which both capture the phenomena of interest and highlight distinct aspects of the framework. The principles themselves are stated in general terms.

From the outset, we emphasize that the applicability of the SB framework *critically depends* on the nature of symmetries present in the learning problem. This dependence is not a limitation but rather a deliberate design principle: the framework is intended to capture the symmetries *jointly* induced by the loss function, the architecture, and the data distribution (see Brutzkus & Globerson (2017); Shamir (2018) for hardness results when one of these components is chosen adversarially). In the absence of symmetries, exact or approximate, the framework remains applicable but, correctly, does not predict, for example, the emergence of a bulk–outlier structure (see (f) in Figure 1). Our contributions may be stated as follows.

- We construct a large ambient space, referred to as the *universal symmetry space*, allowing us to capture symmetries jointly induced by the loss, network architecture, and data. The framework allows for a systematic characterization of the invariance properties of the expected loss associated with fitting three-layer networks under the squared loss. Building on Arjevani (2024), we show that critical points obtained by stochastic gradient descent (SGD) break the symmetry of the expected loss, though only in a limited manner (see Section 5 for precise statements).

- We employ representation-theoretic techniques to estimate the Hessian spectrum at SB points in high dimensions using only a fixed, small subset of Hessian entries (see, e.g., Figure 1, which depicts

spectra for models with over $10^8$ parameters). The analysis reveals significant curvature differences between local and global minima, contrary to the analogous two-layer setting, suggesting that the flat minima conjecture may depend on network depth.

- Using group-theoretic methods, we prove that SB forces the Hessian and the GN matrix at critical points to lie in fixed low-dimensional spaces and to exhibit eigenvalues with multiplicities $\Theta(d^2)$, $\Theta(d)$, and $\Theta(1)$ (Theorem 1). These predictions are corroborate by empirical evidence, providing a unified theoretical account of the bulk-outlier phenomenon reported in the literature LeCun et al. (2012); Sagun et al. (2016; 2017); Ghorbani et al. (2019); Papyan (2018).

- As indicated earlier, the gradient has been observed during optimization to align with low-dimensional subspaces Gur-Ari et al. (2018); Ghorbani et al. (2019), corresponding to Hessian eigenspaces associated with large eigenvalues. Our findings instead identify the symmetry space of the convergence point, being SB, as the main subspace capturing the gradient's dominant directions—overlapping with, but not identical to, leading eigenspaces. Focusing on the end of training, we propose interpreting the collection of symmetry spaces associated with SB critical points as a form of implicit bias, and derive tighter generalization bounds conditioned on this perspective.

- While existing approaches, including those indicated earlier and additional methods from statistical physics Goldt et al. (2019b); Aubin et al. (2019); Goldt et al. (2019a); Oostwal et al. (2021); Mei et al. (2018); Pham & Nguyen (2021b;a); Nguyen (2019), optimal transport Chizat & Bach (2018); Chizat (2021), and NTK theory Jacot et al. (2018); Li et al. (2020b); Hanin & Nica (2019) have yielded valuable insights into the loss landscape, they do so under restrictive parameter regimes that strongly simplify the model's nonconvex nature, for example, by assuming *infinitely wide* hidden or input layers. By contrast, our framework addresses the loss landscape directly in the natural regime, where both the input dimension and the number of neurons are *arbitrary* yet *finite*.

The paper is organized as follows. To carefully account for all relevant symmetries, we begin with a detailed formulation of the SB framework. We then present our main results, which relate SB to the phenomena introduced earlier. To maintain focus on the main theme, the preliminary material is kept minimal at this stage, and certain results are stated informally, with references to their formal versions provided later in the manuscript. Technical sections introducing additional ideas and methods, together with the necessary background, are deferred to the appendix.

## 2 SYMMETRY BREAKING CRITICAL POINTS

We consider a learning problem specified by: i. a class of models $N : \mathbb{X} \times \mathbb{W} \to \mathbb{Y}$, parameterized by weights $W$ in the weight space $\mathbb{W}$, which is endowed with an inner product. Each model maps the input space $\mathbb{X}$ to the output space $\mathbb{Y}$, with the product $\mathbb{X} \times \mathbb{Y}$ being a measurable space equipped with a joint $\sigma$-algebra $\mathcal{B}_{\mathbb{X} \times \mathbb{Y}}$. ii. a loss function mapping $\mathbb{Y} \times \mathbb{Y}$ to $\mathbb{R}$. Given a probability measure $\mathcal{D}$ on $(\mathbb{X} \times \mathbb{Y}, \mathcal{B}_{\mathbb{X} \times \mathbb{Y}})$, the *expected loss* with respect to $\mathcal{D}$ is defined as $\mathcal{L}(W) \coloneqq \mathbb{E}_{(\mathbf{x},y) \sim \mathcal{D}}[\ell(N(\mathbf{x}; W), y)]$. The GN matrix, introduced earlier, is $H_{\mathrm{GN}}(W) \coloneqq 2\mathbb{E}_{\mathbf{x} \sim \mathcal{D}}[\mathrm{D}_W N(W; \mathbf{x})^\top \mathrm{D}_W N(W; \mathbf{x})]$ where the adjoint is taken with respect to the inner product on $\mathbb{W}$. Here and throughout, all relevant quantities are assumed to be integrable with respect to $\mathcal{D}$.

We begin by constructing the *universal symmetry space* $\mathbb{U} = \mathbb{W} \times \mathbb{X} \times \mathbb{Y}$, so as to simultaneously accommodates all symmetries of the weight, input, and output spaces, where symmetries are regarded as transformations $g$ on $\mathbb{U}$, referred to as *endomorphisms*, that preserve each factor: $g(\mathbb{W}) \subseteq \mathbb{W}$, $g(\mathbb{X}) \subseteq \mathbb{X}$, and $g(\mathbb{Y}) \subseteq \mathbb{Y}$. Thus, we often write, for example, $gW$ in place of $g|_{\mathbb{W}}W$, with the domain restriction understood but not explicitly indicated. The set of all such transformations forms a semigroup, which we denote by $\mathrm{End}(\mathbb{U})$. Additional conditions are typically imposed on endomorphisms, such as linearity, invertibility, or affinity, according to the category of morphisms in which we work.

We distinguish two fundamental types of symmetries. The first, *structural symmetries*, denoted by $\mathrm{End}(\mathbb{U}; N, \ell) \leq \mathrm{End}(\mathbb{U})$ (where $\leq$ denotes inclusion as a subsemigroup), are intrinsic to the choice of loss and network architecture, and are defined by requiring $\ell(N(\mathbf{x}; W), y) = \ell(N(g\mathbf{x}; gW), gy)$ for all $(W, \mathbf{x}, y) \in \mathbb{U}$. We may, and shall, consider structural symmetries on subsets of $\mathbb{U}$. The second type, *distributional symmetries*, which we denote by $\mathrm{End}(\mathcal{D})$, capture invariances of the data distribution $\mathcal{D}$ and consist of all transformations $g$ on $\mathbb{X} \times \mathbb{Y}$ such that $\mathcal{D}(g^{-1}(A)) = \mathcal{D}(A)$ for all measurable sets $A \in \mathcal{B}_{\mathbb{X} \times \mathbb{Y}}$. Throughout, we regard transformations in $\mathrm{End}(\mathcal{D})$ as acting trivially on

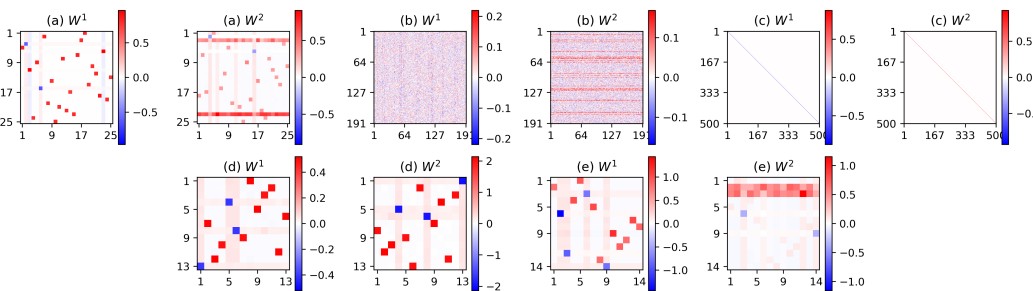

Figure 2: Weight matrices of representative critical points of (2.3). (a,b) Obtained via SGD, with (b) included to illustrate that symmetry breaking, while present, may not be visually apparent in high dimensions (isotropy groups are approximately $S_{21} \times S_1^4$ and $S_{168} \times S_1^{23}$, respectively). (c) A non-global local minimum, unstable and avoided under He initialization (*hidden* in Arjevani (2023)), but detectable by forcibly breaking symmetry (here illustrated with a deliberately simple structure yielding isotropy group $S_{500}$). (d) A local minimum of (2.3) with the underlying distribution replaced by a mixture of two Gaussians. (e) A local minimum of (2.3) with the target network being $2I_7 \oplus 20I_7$. The patterns reflect the breaking of symmetries *jointly* induced by the loss function, the network architecture, and the data distribution.

$\mathbb{W}$, and so have $\mathrm{End}(\mathcal{D})$ canonically embedded in $\mathrm{End}(\mathbb{U})$. (A third type of symmetry, involving the optimization algorithm itself, is treated elsewhere.) If $g \in \mathrm{End}(\mathbb{U}; N, \ell)$ and $h \in \mathrm{End}(\mathcal{D})$ are such that $g \circ h$ acts trivially on $\mathbb{X} \times \mathbb{Y}$, then $\mathcal{L}(gW) = \mathbb{E}[\ell(N(\mathbf{x}; gW), y)] = \mathbb{E}[\ell(N((g \circ h)\mathbf{x}; gW), (g \circ h)y)] = \mathbb{E}[\ell(N(h\mathbf{x}; W), hy)] = \mathbb{E}[\ell(N(\mathbf{x}; W), y)] = \mathcal{L}(W)$. Accordingly, given subsemigroups of $\mathrm{End}(\mathbb{U}; \mathcal{N}, \ell)$ and $\mathrm{End}(\mathcal{D})$, we define the *induced invariance semigroup* to be the set, indeed semigroup, of all such elements.

We illustrate these concepts with respect to the two three-layer networks considered in our setting, mapping from $\mathbb{X} = \mathbb{R}^d$ to $\mathbb{Y} = \mathbb{R}$, with and without residual connections. Although inclusion of bias is also addressed in the paper, see Table 1, we omit it from the present discussion for simplicity, as its analysis is straightforward. Specifically, we consider the models:

$$\bar{N}(\mathbf{x}; W) := \mathbf{w}^{(3)}\sigma(W^{(2)}\sigma(W^{(1)}\mathbf{x})), \quad N(\mathbf{x}; W) := \mathbf{1}_d^\top \sigma(W^{(2)}\sigma(W^{(1)}\mathbf{x}) + \mathbf{x}), \qquad (2.1)$$

where $\sigma$ denotes the ReLU activation and $\mathbf{1}_d$ the $d$-dimensional all-ones vector. For $\bar{N}$, the parameter tuple $W = (W^{(1)}, W^{(2)}, \mathbf{w}^{(3)})$ lies in the weight space $\mathbb{W} := M(d_2, d) \oplus M(d_3, d_2) \oplus M(d_3, 1)$, defined mutatis mutandis for $N$, and $M(d', d'')$ denotes the space of all $d' \times d''$ matrices. Throughout, we use the squared loss $\ell(z) = z^2$. Structural symmetries include permutations of weights connecting adjacent layers (including the input and output layer), giving rise to actions of $S_d$, the symmetry group on $d$ symbols. If $P_\pi$ denotes the permutation matrix corresponding to $\pi \in S_d$, these actions may be given explicitly for $N$ by mapping $(W^{(1)}, W^{(2)}, \mathbf{x}, y) \in \mathbb{U}$ to $(W^{(1)}P_\pi^\top, W^{(2)}, P_\pi\mathbf{x}, y), \pi \in S_d$, $(P_\pi W^{(1)}, W^{(2)}P_\pi^\top, \mathbf{x}, y), \pi \in S_{d_2}$ and $(W^{(1)}, P_\pi W^{(2)}, P_\pi\mathbf{x}, y), \sigma \in S_d$, where the last transformation is instead $(W^{(1)}, P_\pi W^{(2)}, \mathbf{w}^{(3)}P_\pi^\top, \mathbf{x}, y), \pi \in S_{d_3}$ for $\bar{N}$. The squared loss induces a $\mathbb{Z}_2$-action on weights and outputs by flipping the signs of $W^{(3)}$ and $y$ simultaneously. The choice of ReLU as the activation gives rise to continuous symmetries parameterized by $\mathbb{R}_{>0}$, acting by inversely rescaling the incoming and outgoing weights of each hidden neuron.

Distributional symmetries, in contrast, are defined relative to the underlying data distribution. We illustrate with the standard multivariate Gaussian distribution $\mathcal{N}(0, I_d)$ on $\mathbb{X}$, letting $y = T_1(\mathbf{x})$ for some $S_d$-invariant function $T_1$. Additional examples addressing less symmetric settings are provided in Figure 2. While $\mathcal{N}(0, I_d)$ is invariant under all orthogonal transformations, joint invariance on $\mathbb{X} \times \mathbb{Y}$ reduces these to the $S_d$-action permuting inputs and acting *trivially* on $\mathbb{Y}$. Overall, the induced invariance group of the expected loss is $\Gamma := \mathbb{R}_{>0}^{d_2} \rtimes S_d^2 \times S_{d_2} \times \mathbb{Z}_2$ with[1] a residual connection and $\bar{\Gamma} := \mathbb{R}_{>0}^{d_2+d_3} \rtimes S_d \times S_{d_2} \times S_{d_3} \times \mathbb{Z}_2$ otherwise, with actions as above restricted to $\mathbb{W}$ (see Arjevani & Field (2019) for analysis of models with one trainable layer).

---

[1] The semidirect product arises because scaling and permutation do not commute; instead, the subgroup $\mathbb{R}_{>0}^d$ is *closed* under permutations (all groups act trivially on $\mathbb{R}_{>0}^d$ except $S_d$, which permutes the scaling factors). Relatedly, we later encounter the wreath product $\wr$, see (Dummit et al., 2004, p. 187).

**Isotropy group.** Henceforth, although the notions introduced apply to semigroups, we restrict to groups for simplicity. Given an invariance group $\Gamma$ of the expected loss, we measure the symmetry of a point $W$ relative to $\Gamma$ by its *isotropy group*, the subgroup of $\Gamma$ that fixes $W$:

$$\Gamma_W := \{g \in \Gamma \mid gW = W\}. \tag{2.2}$$

In general, a function being invariant $\Gamma$ does not imply that isotropy groups of critical points are large subgroups of $\Gamma$: there exists $\Gamma$-invariant $C^\infty$ functions with global minima having *trivial* isotropy, i.e., $\Gamma_W = \{e\}$, Scheurle & Walcher (2015) . However, by the mechanism introduced in Arjevani (2024), if a critical point $C$ with isotropy group $\Gamma_C$ exists (notably, by virtue of equivariance requirements), then *generically*—with respect to a suitable function topology (see op. cit.)—we expect critical points $W$ that *break the symmetry* of $C$, in the sense that $\Gamma_W$ is, or contains, a large subgroup of $\Gamma_C$. *Large* is defined relative to context. The mechanism employs so-called *tangency* arcs bifurcating from a symmetric critical point while retaining part of its isotropy group (by o-minimal methods, such arcs also determine whether a critical point is minimum, maximum, or saddle; see Section 4.5 in op. cit. and Arjevani (2023)). Critical points along these arcs inherit the symmetry. The argument applies recursively at each newly identified point; See Sections D.1 and A and for examples with the hyperoctahedral group and a tensor decomposition problem, respectively; tangency arcs for the latter appear in Table 2.

To demonstrate the presence of SB critical points predicted by the mechanism, we henceforth fix a concrete setting by restricting to the network $N$ in (2.1), taking all layers to have width $d$, and assuming that labels are generated by a target network as follows:

$$\mathcal{L}(W) = \mathbb{E}_{\mathbf{x} \sim \mathcal{N}(0, I_d)} \left[ (N(\mathbf{x}; W) - T(\mathbf{x}))^2 \right], \quad T(\mathbf{x}) := N(\mathbf{x}; (I_d, \mathbf{1}_d)). \tag{2.3}$$

Since $(I_d, \mathbf{1}_d) = (P_\pi I_d P_\pi^\top, \mathbf{1}_d P_\pi^\top)$ for all $\pi \in S_d$, it follows that $\Gamma_T := \Gamma_{(I_d, \mathbf{1}_d)}$ is isomorphic to $S_d$, and likewise $\bar{\Gamma}_{(I_d, I_d, \mathbf{1}_d)}$. We therefore expect critical points $W$ with large isotropy subgroups of $S_d$ (acting as above). Consistent with the SB mechanism, our experiments indicate that the critical points identified by stochastic gradient descent ( are SB, in that their isotropy groups form large subgroups of $\Gamma_T$, see Figure 2, which also includes further examples with other distributions and targets.

## 3 STRUCTURE AND SPECTRUM OF HESSIAN AND GAUSS–NEWTON MATRICES

As demonstrated by Figure 2, at critical points $C$ that do not break the symmetry of $T$ (2.3) *too much*, in the sense that $\Gamma_C$ is a large subgroup of $\Gamma_T \cong S_d$, the weights exhibit a distinct structure. The precise meaning of *large* will be specified in Section 5; roughly, it refers to the case where the index $|\Gamma_T|/|\Gamma_C|$ grows polynomially with $d$. More is true beyond the distinct structure of the weights.

**Theorem 1** (Informal). *At critical points of the loss $\mathcal{L}$ that do not break the symmetry of $T$ too much, the Hessian and Gauss–Newton matrices:*
*1. lie in the same fixed low-dimensional subspace;*
*2. have spectra with $\Theta(1)$ eigenvalues of multiplicity $\Theta(d^2)$, $O(d)$ eigenvalues of multiplicity $\Theta(d)$, and the remaining eigenvalues each with multiplicity $\Theta(1)$.*
*Both assertions remain valid when the Hessian and Gauss–Newton matrices are computed with respect to a specific layer, with the corresponding modifications applying to the third $d \times 1$ layer.*

The formal statement of the theorem is given in Section 5 (cf. (Arjevani, 2024, Section 3) for higher-order analogs), whereas the full proof, which relies on methods from permutation group theory Aschbacher & Scott (1985); Cameron (1999) and the analysis of the associated *centralizer algebra* and *isotypic decomposition* Serre et al. (1977); Fulton & Harris (1991), is deferred to Section B. Here we ask: does this theorem, purely group-theoretic in nature, have observable consequences in practice? In particular, does the predicted structure of the eigenspectrum manifest in empirical observations? In Figure 1, we display the Hessian eigenspectra of several critical points. The results show strong agreement with the theoretical predictions, as detailed in the figure caption. To the best of our knowledge, the present work offers the first comprehensive theoretical explanation of the bulk-outlier eigenspectra in our settings.

**Continuous symmetries.** Continuous symmetries of the loss $\mathcal{L}$ (here, Lie) enforce vanishing Hessian and GN eigenvalues along directions tangent to the group orbits. Thus, for $\mathcal{L}$, $d_2$ out of the $d_1 d_2 + d_2 d_3$ eigenvalues must vanish on account of the scaling factor $\mathbb{R}_{>0}^{d_2}$ in the induced invariance

group $\Gamma$. More generally, the number of eigenvalues that vanish due to structural symmetries given by reciprocal positive scaling Arjevani & Field (2019); Zhao et al. (2022), grows only linearly with the layer sizes, in contrast to the quadratic growth of the total eigenvalue count. Since in practice the bulk of the spectrum is overwhelmingly large, this effect by itself seems insufficient to account for the observed phenomenon. By contrast, the bulk arising from the $\Theta(d^2)$-multiplicity eigenvalues determined by Theorem 1 does not necessarily vanish but rather emerges naturally from the presence of high-degree *irreducible representations* (see Section B), in agreement with the bulk size observed empirically. For the network $\bar{N}$ (2.1), however, structural symmetries *localized* can enforce the vanishing of $\Theta(d^2)$ eigenvalues—although, unlike before, here the bulk sits exactly at zero. For instance, for all matrices $A$ in the positive orthant, one has $\bar{N}(\cdot; I_d, UA, \mathbf{1}_d) = \bar{N}(\cdot; I_d, A, \mathbf{1}_d)$ for any $U$ in the real algebraic semigroup of left-stochastic matrices (independently of the data distribution). Such structural symmetries constitute a fundamental aspect of the framework and demonstrate why identifying SB weight patterns is not always straightforward (with high dimensionality presenting additional challenges, (b) of Figure 2).

Lie group actions also entail conservation laws for *gradient flow*, in the spirit of Noether's laws. These give rise, for example, to the auto-balancing property Du et al. (2018), as established in (Arjevani & Field, 2019, Section 3), and see more generally Kunin et al. (2020); Marcotte et al. (2023; 2024). While such effects clarify aspects of optimization by identifying quantities preserved along the trajectories, they do not account for phenomena involving substantial shifts, such as those examined next.

## 4 OPTIMIZATION HAPPENS MOSTLY TANGENTIALLY TO FIXED POINT SPACES

The collection of all points, critical or otherwise, that remain invariant under a subgroup $G$ of the invariance group $\Gamma$ (see Section 2) forms a vector space, known as the *fixed point space*,

$$\mathbb{W}^G := \{W \in \mathbb{W} \mid \pi W = W \text{ for all } \pi \in G\}. \tag{4.4}$$

When $G$ acts orthogonally, as we assume henceforth, the gradient behaves particularly well with respect to this space: if $W \in \mathbb{W}^G$, then $\nabla\mathcal{L}(W) \in \mathbb{W}^G$ as well, a direct consequence of the *equivariance* of the gradient map, see Section B. Therefore, convergence to a critical point with isotropy group $\Gamma_W$ implies that the gradient component *transverse* to $\mathbb{W}^{\Gamma_W}$ eventually vanishes. How, then, do the tangential and transverse components of the gradient, relative to $\mathbb{W}^{\Gamma}_W$ at the SB points considered in Theorem 1, compare in magnitude *during* optimization? For typical step sizes, we consistently observe that, although $\mathbb{W}^{\Gamma_W}$ is of negligible dimension compared with the ambient space (dimensions $o(d)$ and $\Theta(d^2)$, respectively), the gradient lies predominantly within $\mathbb{W}^{\Gamma_W}$ already at the early stages of the optimization (see Figure 3). Moreover, by standard perturbation-theoretic arguments, the structure and eigenspectra of the Hessian and GN matrices dictated by

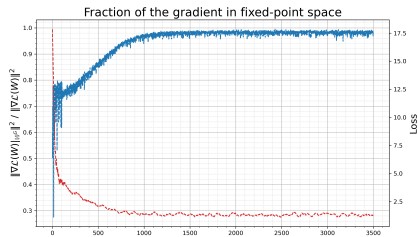

Figure 3: Loss and fraction of the gradient in the fixed point space of the convergence point, illustrating that optimization is largely tangential to this space. Learning rate 0.1 as in Gur-Ari et al. (2018).

Theorem 1 are expected to emerge not only upon convergence but also early in training, when the gradient aligns with $\mathbb{W}^{\Gamma}_W$, yielding, for example, clusters of eigenvalues forming around discrete eigenvalues at the fixed points. Another consequence of the gradient restricting to fixed point spaces is that, modulo finite-precision effects, the symmetry of trajectories of gradient descent, whether stochastic or deterministic, can only increase Arjevani & Field (2019), suggesting a form of *incremental learning* (see Kalimeris et al. (2019); Nakkiran et al. (2019); Pesme & Flammarion (2023)).

**Gradient Descent Happens in a Tiny Subspace.** As noted, it has been consistently observed Gur-Ari et al. (2018); Ghorbani et al. (2019) that: (i) the gradient concentrates in low-dimensional subspaces early in training, and (ii) these subspaces remain largely decoupled from the bulk of eigenvalues throughout optimization. The low-dimensional subspaces were identified with the leading eigenspaces of the Hessian. Despite attempts to account for this phenomenon Gur-Ari et al. (2018); Ghorbani et al. (2019), the mechanism driving this behavior remained unclear. Within the

SB framework, the phenomenon admit a natural interpretation. Gradient dynamics converge to fixed point subspaces and, under the training parameters used in Gur-Ari et al. (2018); Ghorbani et al. (2019), evolve predominantly tangentially to the them, as illustrated in Figure 3. The large isotropy group characterizing SB points implies that the associated subspaces are typically of low dimension. Moreover, with the fixed-point subspace determined by the isotropy group of the convergence point, tangential and transverse components remain decoupled throughout. Finally, we note that previous studies of how curvature evolve as a function of the step size have identified a sharp transition early in training, e.g., Jastrzebski et al. (2020); Cohen et al. (2021). Our preliminary experiments hint at a possible link between the shift from progressive sharpening to edge-of-stability and the alignment of the gradient with the fixed-point space, though this remains to be clarified.

**Parameterized families of the loss function: eigenvalues asymptotics.** It is often instructive to study how properties of a critical point change in a parameterized family of functions that includes the loss function itself. For instance, one may replace the ReLU activation with a Leaky ReLU and vary the leakiness parameter, as in (Arjevani & Field, 2021a, indirect method), where such paths are constructed between critical points lying in the same fixed-point space. Another example is increasing the layer width to examine, for example, whether a minimum becomes a saddle point or vice versa. Since the gradient restricts to fixed-point spaces, $d$ may be varied *continuously* (Arjevani & Field, 2021a, direct method), with stability typically shifting at fractional values of $d$ Arjevani & Field (2021b); Arjevani (2024). Here, we increase $d, d_2$ and $d_3$ simultaneously for both networks considered (2.1) and, using the fact that the Hessian lies in a fixed-dimensional space independent of $d$ (see Theorem 1), we derive the growth rates of the eigenvalues, see Table 1.

| Eigenvalue multiplicity (of $\Theta(d^2)$) | 1 | $\Theta(d)$ | $\Theta(d^2)$ |
|---|---|---|---|
| Growth rate | $\Theta(d^2)$ | $\Theta(d)$ | $\Theta(1)$ |

Table 1: Asymptotic Hessian spectra at global minima of (2.3) exhibit $\Theta(d^2)$ growth rate, in sharp contrast to the two-layer case (Arjevani & Field, 2021b, Theorem 2), where the maximal growth rate is $\Theta(d)$. This indicates a rapidly increasing condition number with $d$, which may impede precise convergence in the final stages of training. The asymptotics hold across all variants, with or without bias or residuals, as well as for the local minima we examined. While it is possible to derive sharper estimates—e.g., $d^2 - 3d + 1$ eigenvalues at $c_1 + o(1)$ with $c = 0.03\ldots$, and one eigenvalue of multiplicity 1 at $0.5d + c + o(1)$, with $c = 0.1\ldots$ (bias) or $-0.8\ldots$ (no bias)—rather than distribution-specific constants, our focus is on *qualitative* properties of the loss landscape.

**Model Complexity via Symmetry Breaking: Implicit Bias and Generalization Bounds.** The following is intended as a high-level conceptual illustration of how SB may lead to tighter generalization bounds. Classical bounds typically scale with the number of parameters and are often vacuous in practice Dziugaite & Roy (2017). Consequently, generalization has been attributed to an *inductive bias*, whether explicit or implicit, motivating for example the development of norm-based bounds Bartlett (1996); Anthony & Bartlett (2009); Golowich et al. (2018); Galanti et al. (2023); Neyshabur et al. (2015). We propose that SB can be interpreted as providing such a bias, structural rather than norm-based, since minima with large isotropy groups relative to an invariance group $\Gamma$ typically lie in subspaces of substantially lower dimension than the full parameter space. To formalize this, we define the $\Gamma$-dimension $\mathfrak{C}(\mathcal{G}; \Gamma)$ of a set of isotropy groups $\mathcal{G}$ as the largest dimension of the corresponding fixed-point spaces. The following bound is then established using standard arguments from statistical learning theory (see Section C for definitions, constants, and proofs).

**Proposition 1** (Notation as above). *The pseudodimension of the class of networks $N$ (2.1) with isotropy groups in $\mathcal{G}$ is $O(\mathfrak{C}(\mathcal{G}; \Gamma) \log(d \log(d)))$.*

As a corollary, we find that for the isotropy groups considered in Theorem 1, this yields an $o(d)$-bound (up to logarithmic factors) rather than $O(d^2)$.

## 5 FORMAL STATEMENT OF THEOREM 1

We next present the formal statement of Theorem 1. The full proof is provided in Section B, while the present section is restricted to several basic definitions needed for its formulation. A linear action of a

group $G$ on $\mathbb{W}$ is called a *representation*. A representation is *irreducible* if the only linear subspaces of $\mathbb{W}$ that are preserved (invariant) under the $G$-action are $\mathbb{W}$ and $\{0\}$. In the general theory, each irreducible representation of $S_d$, the isotropy group of $T$ (2.3), is associated with a partition $\lambda$ of the set $[d]$, which we denote by $V_\lambda$. Representations of finite groups can always be expressed as sums of irreducible representations. A map $F : \mathbb{W} \to \mathbb{W}$ is called $G$-equivariant if $F(gW) = gF(W)$ for all $g \in G$, $W \in \mathbb{W}$. If $F$ is linear and equivariant, we call $F$ a $G$-*map*. Both the Hessian and the GN matrices are $G$-maps for any subgroup $G$ of the invariance group of $\mathcal{L}$, see Section B. As we further describe in Section B, the structure of the spectrum of a $G$-map is determined by the irreducible representations constituting $\mathbb{W}$.

The proof of Theorem 1 relies on the O'Nan–Scott theorem Aschbacher & Scott (1985); Cameron (1999), which help establish a connection between the size of a group and the nature of its action. This relationship is then used to deduce lower bounds on the corresponding isotypic decomposition, the latter expressed in terms of irreducible representations of $S_d$, which may be computed either through stabilization of $S_d$-representations (Arjevani, 2024, Section 5.3) or explicitly, as in Section B.

**Theorem 1** (Formal)**.** *Let $d := d_1 = d_2 = d_3 \geq 7$, and let $G = \Gamma_T \cong S_d$, as defined in Section 2. If $[S_d : G] < \binom{d}{p} < \frac{1}{2}\binom{d}{\lfloor d/2 \rfloor}$ for some $0 \leq p \leq d/2$, then any $G$-map has: two eigenvalues of multiplicity $(d-p-1)(d-p-2)/2$, corresponding to $V_{(d-p-2,1,1)}$; two of multiplicity $(d-p)(d-p-3)/2$, from $V_{(d-p-2,2)}$; $4p+7$ of multiplicity $d-p-1$, from $V_{(d-p-1,1)}$; and $2p^2+5p+5$ of multiplicity 1, from $V_{(d)}$. When bias terms are included, there are two additional eigenvalues of multiplicity $d-p-1$ from $V_{(d-p-1,1)}$, and $2p+2$ more of multiplicity 1, from $V_{(d)}$. Generically, the eigenvalues are pairwise distinct, apart from their prescribed multiplicities.*

Empirically, we observe that although smaller isotropy groups, such as those refining $S_d \wr S_2$ exist, $p = o(d)$ already accounts for the groups detected numerically. We emphasize that in contrast to prior work, which focused exclusively on specific groups of the form $S_{d-p} \times S_p$, $p = 0, 1, 2, 3$, Arjevani & Field (2021b), our result applies to a class of subgroups defined by their size, not structure.

Referring to the structure of the Hessian, identical to that of the GN matrix, and the computation of the eigenspectra, the procedure outlined in Section B relies on evaluating Hessian-vector products which may become computationally prohibitive in extremely high-dimensional regimes required for our analysis of growth rates. To overcome this, we exploit further stabilization properties of $S_d$-representations (Arjevani, 2024, Section 5.3), this time applied to the space in which the Hessian and the GN matrices reside, which for example, at the global minimum determined by $T$, is given by,

$$\text{Sym}^2(\mathbb{R}^d \otimes \mathbb{R}^d \oplus \mathbb{R}^d \otimes \mathbb{R}^d) \approx 3V_{(d-4,1,1,1,1)} + 5V_{(d-4,2,1,1)} + 6V_{(d-4,2,2)} + 5V_{(d-4,3,1)} + 3V_{(d-4,4)}$$
$$+ 18V_{(d-3,1,1,1)} + 40V_{(d-3,2,1)} + 22V_{(d-3,3)} + 57V_{(d-2,1,1)} + 69V_{(d-2,2)} + 79V_{(d-1,1)} + 37V_{(d)}.$$

For $d \geq 8$, the Hessian structure stabilizes and is fully described by 37 parameters, see Figure 1, enabling us to recover the spectrum from 37 Hessian entries without using Hessian-vector products; See Section B for details of the expressions defining the corresponding coefficient matrices.

## 6 Concluding remarks

In this work, we have demonstrated that, in the presence of symmetries jointly induced by the loss, the network architecture, and the data distribution, SB emerges as a unifying mechanism shaping the geometry of loss landscapes. The motivation for analyzing the three-layer case, and the guidance for formulating and identifying SB, derives from recent theoretical results Arjevani (2024), which connect SB to invariance properties of the loss function and to the presence of highly symmetric critical points. Once identified, SB phenomena help elucidate key aspects of DL, such as the bulk–outlier spectra of the Hessian and GN matrices and the concentration of gradients in low-dimensional spaces, and further enable the exploration of curvature-related conjectures in very high dimensions.

More broadly, our results support the view that SB may serve as a fundamental principle in the theory of DL. While the specific manifestations of symmetry may vary across domains—for example, local translations in images—the overarching message is that meaningful symmetries tend to arise naturally, and these, in turn, give rise to SB phenomena. Extending results and methods to full-scale architectures is a subject of our ongoing work.

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

## A  SYMMETRIC TENSOR DECOMPOSITION

Consider the problem of decomposing a real symmetric tensor into a sum of rank-one terms **??**. Standard approaches involve solving the following nonconvex optimization problem associated with an order-$n$ symmetric tensor $A$ on $\mathbb{R}^d$:

$$\min_{\substack{W \in M(k,d), \\ \boldsymbol{\alpha} \in \mathbb{R}^k}} \mathcal{L}_A(W, \boldsymbol{\alpha}) \coloneqq \left\| \sum_{i=1}^{k} \alpha_i \mathbf{w}_i^{\otimes n} - A \right\|^2. \tag{A.5}$$

The loss function $\mathcal{L}_A$ is always invariant under permutations of the rows of $W$. Invariance under permutation of the columns of $W$ reflects the symmetry of the target tensor $A$. For example, assume $k = n = 3$. Since $n$ is odd, each $\alpha_i$ can be absorbed into the corresponding $\mathbf{w}_i$, allowing us to set all $\alpha_i = 1$. If we take $A_\mathbf{e} \coloneqq \sum_{i=1}^{d} \mathbf{e}_i^{\otimes 3}$, where $\mathbf{e}_i$ denotes the $i$th standard basis vector, then $\mathcal{L}_A$ is also invariant under permutations of the columns of $W$, leading to an overall $S_d \times S_d$-invariance with respect to the described group action. By construction, $A$ is a global minimizer of the loss function. Its isotropy group is given by $\Delta S_d \subset S_d \times S_d$, where $\Delta$ as indicated in the main text denotes the diagonal embedding of a group $G$ into $G \times G$. The origin is a critical point of the loss function, indeed, a saddle, with full isotropy group $S_d \times S_d$. Tangency arcs connecting critical points of (A.5) exhibiting large isotropy groups are described in Table 2.

| Tangency arc | Source | | | Target | | |
|---|---|---|---|---|---|---|
| $0_2 \oplus t\mathbf{1}_{d-2}\mathbf{1}_d^\top$ | 0 | Saddle | $S_d \times S_d$ | $0_2 \oplus \Theta(1)\mathbf{1}_{d-2}\mathbf{1}_d^\top$ | Saddle | $S_2 \times S_{d-2}$ |
| $tI_d$ | 0 | Saddle | $S_d \times S_d$ | $I_d$ | Global minimum | $\Delta S_d$ |
| $tI_{d-1} \oplus [0]$ | 0 | Saddle | $S_d \times S_d$ | $I_{d-1} \oplus [0]$ | Saddle | $\Delta S_{d-1}$ |
| $I_{d-1} \oplus [t]$ | $I_{d-1} \oplus [0]$ | Saddle | $\Delta S_{d-1}$ | $I_d$ | Global minimum | $\Delta S_d$ |
| $I_{d-2} \oplus \left[\begin{smallmatrix} a(t) & 0 \\ t & 0 \end{smallmatrix}\right]$, $a(t) = \sum_{m=0}^{\infty} \binom{1/3}{m}(-1)^m t^{3m}$ | $I_{d-1} \oplus [0]$ | Saddle | $\Delta S_{d-1}$ | $I_{d-2} \oplus \left[\begin{smallmatrix} 0 & 0 \\ 1 & 0 \end{smallmatrix}\right]$ | Saddle | $\Delta S_{d-2}$ |

Table 2: Tangency arcs associated to a symmetric tensor decomposition problem, Section A.

| Extremal Character | Isomorphism Type | Weight Matrix | Eigenvalues ($6 \times 7 = 42$) | |
|---|---|---|---|---|
| | | | Mult. | # |
| Global | $S_6$ |  | 1 | 3 |
| | | | 5 | 4 |
| | | | 9 | 1 |
| | | | 10 | 1 |
| Spurious | $S_4 \times S_2$ |  | 1 | 13 |
| | | | 2 | 1 |
| | | | 3 | 9 |
| Spurious | $S_3 \wr S_2$ |  | 1 | 8 |
| | | | 2 | 1 |
| | | | 4 | 8 |

Table 3: Isotropy types and the number of critical points associated with fitting (A.6), along with the corresponding structure of the Hessian as dictated by the associated isotypic decomposition.

## A.1 TWO-LAYER NETWORK WITH BIAS

Consider a network with two layers: the first one is fully-connected with bias and the second layer is also fully connected but the weights are fixed and set to one. Formally,

$$N(\mathbf{x}; W^{(1)}) := \mathbf{1}^\top W^{(1)} \begin{pmatrix} \mathbf{x} \\ 1 \end{pmatrix}. \tag{A.6}$$

where $W_1 \in M(d, d+1)$. Clearly, $N$ is invariant under permutations of the rows of $W^{(1)}$. Under the squared loss, fitting with respect to labels produced by the target network $T' := N(\mathbf{x}; [I_d; \mathbf{0}])$ with inputs distributed by the standard multivariate distribution we have that $\mathcal{L}_1(W^{(1)}) := \mathbb{E}_{\mathbf{x} \sim \mathcal{N}(0, I_d)}[(N(\mathbf{x}; W^{(1)}) - T'(\mathbf{x}))^2]$ is invariant under permutation of the rows and the $d$-leftmost columns. The isotropy group at the global minimum $W^{(1)} = [I_d; 0]$ is $\Delta S_d \subseteq S_d \times S_{d+1}$ (properly embedded). The isotypic decomposition is $3V_{(d)} + 4V_{(d-1,1)} + V_{(d-2,1,1)} + V_{(d-2,2)}$ ($d$ sufficiently large), implying that the Hessian has at most 7 distinct eigenvalues.

## B INDEX TO SPECTRUM, O'NAN–SCOTT THEOREM AND GROUP REPRESENTATION

This section supplements the technical background of the main text, highlighting only the material essential for our results. Some parts are intentionally repeated for continuity.

### GROUP ACTION

Some familiarity with group theory is assumed, including: the automorphism group $\mathrm{Aut}(G)$ of a group $G$; the inner $\mathrm{Inn}(G)$ and outer $\mathrm{Out}(G)$ automorphism groups; $Z(G)$ the center of a group; the general linear group $\mathrm{GL}_d(K) = \mathrm{Aut}(K^d)$ of degree $d$ over a field $K$; and the related affine group $\mathrm{AGL}_d(K) = K^d \rtimes \mathrm{GL}_d(K)$ and projective group $\mathrm{PGL}_d(K) = \mathrm{GL}_d(K)/Z(\mathrm{GL}_d(K))$, along with their *special* versions, ASL and PSL, respectively. For finite fields, we write $p^m$ instead of $K$ to denote the field of order $p^m$, unique up to isomorphism. Our exposition focuses entirely on permutation groups, their actions, and their linear (permutation) representations.

The *symmetric group* $\mathrm{Sym}(X)$ is the group of all bijections from $X$ to itself. We write $S_d \doteq \mathrm{Sym}([d])$ and $A_d$ for the group of even permutations. A *permutation group* $H$ is a subgroup of $S_d$.

A *group action* of $G$ on $X$ is a homomorphism $\varphi$ from $G$ to $\mathrm{Sym}(X)$. When the set being acted upon is a group $X = N$, we may consider $\mathrm{Aut}(N) \leq \mathrm{Sym}(X)$ to respect the group structure, yielding the usual semidirect products $N \rtimes_\varphi G$ appearing in the definition of $\mathrm{AGL}_d(K)$, and the direct product $N \times G$ for $\varphi$ trivial. If $G \subseteq S_d$ is a permutation group, the *wreath product* $N \wr G$ is defined by letting $G$ act on the $d$-fold direct product $N^d$ by permuting the tuple subscripts.

We say that $G$ acts *transitively* on a set $X$ if for any two points $a, b \in X$, there is an element of $g \in G$ mapping $a$ to $b$ by its action. More generally, $G$ is *$m$-transitive* if it acts transitively on the set of if all $m$-tuples of distinct elements in $X$. Clearly, $S_d$ is $d$-transitive. The *alternating group* $A_d \subseteq S_d$ of even permutations may be shown to be $(d-2)$-transitive.

MAXIMAL SUBGROUPS OF $S_d$ AND O'NAN–SCOTT THEOREM

Some subgroups of $S_d$ are obvious. For instance, Young subgroups $S_{\lambda_1} \times \cdots S_{\lambda_m}$ associated to integer partitions $(\lambda_1, \ldots, \lambda_m)$ of $d$. To study arbitrary subgroups $G \subseteq S_d$, it is often convenient to first classify the maximal subgroup containing $G$ by the nature of its action on $[d]$. These are classified by O'Nan–Scott theorem asserting (in its original form Aschbacher & Scott (1985); Cameron (1999)) that if $H \neq A_d$ is a maximal subgroup of $S_d$ then $H$ is one of the following:

(a) intransitive, $S_p \times S_q$, $p + q = d, p \neq q$

(b) imprimitive, $S_p \wr S_q$, $pq = d, p, q > 1$

(c) primitive wreath product, $S_p \wr S_q$, $p^q = d, p \geq 5, q > 1$

(d) affine, $\mathrm{AGL}_m(p), p^m = d, p$ prime;

(e) diagonal, $T^m(\mathrm{Out}(T) \times S_m)$, with $T$ a nonabelian simple group, $m \geq 2$ and $d = |T|^{m-1}$

(f) almost simple, $T \trianglelefteq G \leq \mathrm{Aut}(T), T \neq A_d$ nonabelian simple.

The groups mentioned are shown to be generally maximal in Liebeck et al. (1987): types (a)-(e) under the natural assumption $HA_d = S_d$, and type (f) with an explicit list of exceptions, for instance, $\mathrm{PSL}(2,7) < \mathrm{AGL}(3,2) < S_8$.

SUBGROUPS OF $S_p \times S_p$

Subgroups of direct products are constructed by a simple rule given by Goursat's lemma. We begin with a general discussion of characterizing the lattice of subgroups of directs products and then restrict to $S_p \times S_p$ .

Goursat's lemma asserts that the subgroups of $G_1 \times G_2$ are in bijection with isomorphisms between sections of $G$ and $H$. More prosaically, given a quintuple $(H_1, N_1, H_2, N_2, f)$ with $N_i \trianglelefteq H_i \leq G$, $i = 1, 2$, and $f : H_1/N_1 \to H_2/N_2$ an isomorphism, define $H_f = \pi^{-1}(\Delta_f)$ where $\Delta_f$ is the graph of $f$ and $\pi : H_1 \times H_2 \to H_1/N_1 \times H_2/N_2$ the quotient map. The construction may be reversed by setting $H_i \doteq \pi_i(G_1 \times G_2)$ and $N_i \doteq \iota_i^{-1}(G_1 \times G_2)$, $\pi_i$ and $\iota_i$ being the canonical projections and injections, and have $N_1 h_1$ mapped to $N_2 h_2$ with for some $h_2$ such that $(h_1, h_2) \in H$.

**Lemma 1.** *[Goursat. Notation and assumption as above] The two constructions above are well-defined and are inverses to each other. Moreover, by a version of the latter, $H \trianglelefteq G_1 \times G_2$ iff $N_i, H_i \trianglelefteq G_i$ and $H_i/N_i \leq Z(G_i/N_i)$.*

A subgroup $H \leq G_1 \times G_2$ is a *subproduct* if $H = H_1 \times H_2$, for some $H_i \subseteq G_i, i = 1, 2$ (corresponding to trivial sections), and it is a *subdirect product* if the projections $\pi_i$ are surjective (despite the similarity, the two are used in the literature). Finally, for short, we will use $\Delta G \subseteq G \times G$ to indicate the graph of the identity automorphism of $G$.

**Example 1.** *The direct product $S_2 \times S_2$ (Klein four group) has four subgroups $\langle (e) \rangle, \langle ((12), e) \rangle$, $\langle (e, (12)) \rangle, \langle ((12), (12)) \rangle$ associated by the construction above to the quintuples $(\langle e \rangle, \langle e \rangle, \langle e \rangle, \langle e \rangle, \mathrm{id})$, $(\langle (12) \rangle, \langle (12) \rangle, \langle e \rangle, \langle e \rangle, \mathrm{id}), (\langle e \rangle, \langle e \rangle, \langle (12) \rangle, \langle (12) \rangle, \mathrm{id}), (\langle (12) \rangle, \langle e \rangle, \langle (12) \rangle, \langle e \rangle, \mathrm{id})$, respectively. The example is as simple as it gets; however, already here, the fourth subgroup illustrates how subgroups that do not correspond to a direct product of subgroups occur for non-trivial sections.*

*In the general case, isomorphic non-trivial sections can admit more than one isomorphism, each associated to a distinct subgroup. Importantly, whether $S_2 \times S_2$ acts transitively as a permutation group depends on its embedding in $S_4$, or equivalently, its action on [4].*

**Example 2.** *The direct product $S_3 \times S_3$ has 60 subgroups, 10 of which are normal. A complete enumeration may be achieved by exhausting all quintuples of sections. Thus, the subgroups of $S_3$ are $\langle (e) \rangle, \langle (12) \rangle, \langle (13) \rangle, \langle (23) \rangle, A_3 = \langle (123) \rangle, S_3$. There are 6 trivial sections obtained by taking the quotient of each subgroup by itself, giving 36 distinct subgroups of $S_3 \times S_3$ in total. These are exactly the subgroups given by direct product of subgroups of $S_3$, 9 of which have $\langle e \rangle, A_3, S_3$ as factors hence are normal.*

*The 4 sections of size 2 are: $\langle (ij) \rangle / \langle e \rangle$, $i \neq j$ and $S_3/A_3$. Each pair admits a unique isomorphism, giving in total 16 distinct subgroups of $S_3 \times S_3$, of which the one associated to $S_3/A_3 \to S_3/A_3$ is the only normal subgroup. There is only one section of size 3, $A_3/\langle e \rangle$ and 2 isomorphisms determined by mapping $(123) \mapsto (123)$ or $(123) \mapsto (132)$ yielding 2 distinct groups (neither is normal, $A_3$ being its own centralizer). Finally, there is one section of size 6, $S_3/\langle e \rangle$, and 6 isomorphisms, each corresponding to an inner automorphism (conjugation).*

The task of enumerating all subgroups of direct products, however unwinding, becomes quite formidable as $d$ increases. For instance, already for for $S_5 \times S_5$ and $S_7 \times S_7$, there are 2976 and 101921 subgroups, respectively (reducing modulo conjugation does not offer much improvement as there are 809 and 53,125 conjugacy classes, respectively). Fortunately, in the context of symmetry breaking, the analysis can often be reduced to characterizing large groups situated high in the lattice of subgroups, whence the following lemma, see (Rose, 1965, Section 2).

**Lemma 2.** *Let $f : G \to H$ be an isomorphism and $\Delta_f$ the graph of $f$. Then the lattice of subgroups of $G \times H$ containing $\Delta_f$ is isomorphic to the lattice of normal subgroups of $G$. In particular, $\Delta_f$ is maximal iff $G$, hence $H$, are simple.*

For instance, $\Delta A_d$ is maximal in $A_d \times A_d$, but $\Delta S_d$ is *not* maximal in $S_d \times S_d$. The reminder of the section is devoted to the latter case. In addition, henceforth, we write $H \leq^m G$ to indicate that $H$ is a maximal subgroup of $G$ (or, when decisiveness is called for, $H <^m G$ if $H$ is further a *proper* subgroup).

If a group $H$ is not a subdirect product then it is contained in the subproduct of its projections, hence equal to it by maximality, and every projection must be maximal in its factor (the converse direction is clear). Otherwise, if $H$ is a subdirect product, then by the lemma above the two sections, $(G_i/N_i, i = 1, 2$, in the notation above) must be simple.

**Proposition 2.** *If $d_2 \geq d_1 \geq 5$ and $H \subseteq S_{d_1} \times S_{d_2}$ is maximal, then either $H$ is a subproduct of a maximal subgroup of one factor (listed in Section B) with the other factor, or $H$ is the group of pairs of permutations with identical signs. Moreover (embeddings implied),*

$$\Delta S_{d_1} \times S_{d_2 - d_1} <^m A_{d_1} \Delta S_{d_1} \times S_{d_2 - d_1} <^m S_{d_1} \times (S_{d_1} \times S_{d_2 - d_1}) \leq^m S_{d_1} \times S_{d_2}. \quad \text{(B.7)}$$

For the special cases where both $d_1, d_2 \leq 4$ (direct product commutes), maximal groups are easy to compute explicitly (and see Example 1 and Example 2). Finally, we remark that for subproducts, the argument may be repeated. The diagonal case is also simple to handle, modulo Section B, as $H \leq^m \Delta S_d$ implies $H = \Delta K$ for some $K \leq^m S_d$.

GROUP REPRESENTATIONS

Let $V$ be a finite dimensional (mostly real, but also complex on occasions) vector space with inner product $\langle \, , \, \rangle$ and norm $\| \; \|$. An $\mathbb{R}$-representation of a finite group $G$ is a pair $(\rho, V)$ of group homomorphism $\rho$ mapping $G$ to $\mathrm{GL}(V)$, $V$ being an Euclidean space. The *degree* of the representation is the dimension of $V$.

If $G$ acts by permuting the elements of some basis elements of $V$, we say that $(\rho, V)$ is a (linear) *permutation representation*—the work exclusively studying permutation representations. If $G$ acts orthogonally on $V$, we say that $(\rho, V)$ is an *orthogonal representation*. Usually, we just say $(\rho, V)$ is a *representation* of $G$, omitting the indication of the underlying field being $\mathbb{R}$ and assume orthogonality. A representation $(\rho_G, V)$ is *trivial* if each element of $G$ acts as the identity on $V$. If $(\rho_G^1, V_1)$ and $(\rho_G^2, V_2)$ are two representations, the direct sum $V_1 \oplus V_2$ and the tensor product $V_1 \otimes V_2$ are also

representations, the latter via $g(\mathbf{v}_1 \otimes \mathbf{v}_2) = g\mathbf{v}_1 \otimes g\mathbf{v}_2$. The $m$th tensor power $V^{\otimes m}$, the exterior powers $\wedge^m V$ and the symmetric powers $\mathrm{Sym}^m(V)$ are representations by the same rule. The dual representation $V^* = \hom(V, \mathbb{R})$ is defined by $g\mathbf{v}^* = \mathbf{v}^* \circ g^{-1}$, $\mathbf{v}^* \in V^*$, making $\hom(V_1, V_2)$ a representation via the identification $\hom(V_1, V_2) = V_1^* \otimes V_2$.

**Example 3.** *The permutation representation $(\rho_{S_d}, \mathbb{R}^d)$ is defined by allowing $S_d$ to freely permute the $d$ entries. Additional natural permutation representations may be constructed by restricting to subgroups of $S_{k \times d}$. Thus, properly embedding $S_k \times S_d$ in $S_{k \times d}$, we obtain the external tensor representation $\mathbb{R}^k \boxtimes \mathbb{R}^d \cong M(k, d)$, which we refer to as $(\rho_{S_k \times S_d}, M(k, d))$, where the first factor $S_k$ permutes rows, and the second factor $S_d$ columns. The representation $(\mathbb{Z}_k \times \mathbb{Z}_d, M(k, d))$ is similarly defined. When $k = d$, we may consider the internal tensor representation $\mathbb{R}^d \otimes \mathbb{R}^d \cong M(d, d)$ with $S_d$ acting diagonally by simultaneously permuting rows and columns, denoted by $(\rho_{S_d}, M(d, d))$ (and similarly for $\mathbb{Z}_d$).*

THE SPECTRUM OF EQUIVARIANT LINEAR ISOMORPHISMS

If $G$ is a subgroup of $\mathrm{O}(d)$, the action on $\mathbb{R}^d$ is called an *orthogonal* representation of $G$ (we often drop the qualifier orthogonal). Denote by $(\mathbb{R}^d, G)$ as necessary. The *degree* of a representation $(V, G)$ is the dimension of $V$ ($V$ will always be a linear subspace of some $\mathbb{R}^n$ with the induced Euclidean inner product). The action of $S_k \times S_d \subset S_{k \times d}$ on $M(k, d)$ is orthogonal with respect to the standard Euclidean inner product on $M(k, d) \approx \mathbb{R}^{k \times d}$ since the action permutes the coordinates of $\mathbb{R}^{k \times d}$ (equivalently, components of $k \times d$ matrices). Given two representations $(V, G)$ and $(W, G)$, a map $A : V \to W$ is called $G$-equivariant if $A(gv) = gA(v)$, for all $g \in G, v \in V$. If $A$ is linear and equivariant, we say $A$ is a *G-map*. Invariant functions naturally provide examples of equivariant maps. Thus the gradient $\nabla \mathcal{L}$ is a $S_k \times S_d$-equivariant self map of $M(k, d)$ and if $W$ is a critical point of $\nabla \mathcal{L}$ with isotropy $G \subset S_k \times S_d$, then $\nabla^2 \mathcal{L}(W) : M(k, d) \to M(k, d)$ is a $G$-map.

A representation $(\mathbb{R}^n, G)$ is *irreducible* if the only linear subspaces of $\mathbb{R}^n$ that are preserved (invariant) by the $G$-action are $\mathbb{R}^n$ and $\{0\}$. Two orthogonal representations $(V, G)$, $(W, G)$ are *isomorphic* (and have the same *isomorphism class*) if there exists a $G$-map $A : V \to W$ which is a linear isomorphism. If $(V, G)$, $(W, G)$ are irreducible but not isomorphic then every $G$-map $A : V \to W$ is zero (as the kernel and the image of a $G$-map are $G$-invariant). If $(V, G)$ is irreducible, then the space $\mathrm{Hom}_G(V, V)$ of $G$-maps (endomorphisms) of $V$ is a real associative division algebra and is isomorphic by a theorem of Frobenius to either $\mathbb{R}, \mathbb{C}$ or $\mathbb{H}$ (the quaternions). The *only* case that will concern us here is when $\mathrm{Hom}_G(V, V) \approx \mathbb{R}$ when we say the representation is *real*.

**Example 4.** *Let $n > 1$. Take the natural (orthogonal) action of $S_n$ on $\mathbb{R}^n$ defined by permuting coordinates. The representation is not irreducible since the subspace $\{(x, x, \cdots, x) \in \mathbb{R}^n \mid x \in \mathbb{R}\}$ isomorphic to $V_{(d)}$ is invariant by the action of $S_n$, as is the hyperplane $\{(x_1, \cdots, x_n) \mid \sum_{i \in [n]} x_i = 0\}$. It is easy to check that the former, also called the trivial representation of $S_n$, and the latter, the standard representation, are irreducible, real, and not isomorphic.*

Every representation $(\mathbb{R}^n, G)$ can be written uniquely, up to order, as an orthogonal direct sum $\oplus_{i \in [m]} V_i$, where each $(V_i, G)$ is an orthogonal direct sum of isomorphic irreducible representations $(V_{ij}, G)$, $j \in [p_i]$, and $(V_{ij}, G)$ is isomorphic to $(V_{i'j'}, G)$ if and only if $i' = i$. The subspaces $V_{ij}$ are *not* uniquely determined if $p_i > 1$. If there are $m$ distinct isomorphism classes $\mathfrak{v}_1, \cdots, \mathfrak{v}_m$ of irreducible representations, then $(\mathbb{R}^n, G)$ may be represented by the sum $p_1 \mathfrak{v}_1 + \cdots + p_m \mathfrak{v}_m$, where $p_i \geq 1$ counts the number of representations with isomorphism class $\mathfrak{v}_i$. Up to order, this sum (that is, the $\mathfrak{v}_i$ and their multiplicities) is uniquely determined by $(\mathbb{R}^n, G)$. This is the *isotypic decomposition* of $(\mathbb{R}^n, G)$ (see Thomas (2004)). The isotypic decomposition is a powerful tool for extracting information about the spectrum of $G$-maps.

If $G = S_d$, then every irreducible representation of $S_d$ is real (Fulton & Harris, 1991, Thm. 4.3). Suppose, as above, that $(\mathbb{R}^n, S_d) = \oplus_{i \in [m]} V_i$ and $A : \mathbb{R}^n \to \mathbb{R}^n$ is an $S_d$-map. Since the induced maps $A_{ii'} : V_i \to V_{i'}$ must be zero if $i \neq i'$, $A$ is uniquely determined by the $S_d$-maps $A_{ii} : V_i \to V_i$, $i \in [m]$. Fix $i$ and choose an $S_d$-representation $(W, S_d)$ in the isomorphism class $\mathfrak{v}_i$. Choose $S_d$-isomorphisms $W \to V_{ij}$, $j \in [p_i]$. Then $A_{ii}$ induces $\overline{A}_{ii} : W^{p_i} \to W^{p_i}$ and so determines a (real) matrix $M_i \in M(p_i, p_i)$ since $\mathrm{Hom}_{S_d}(W, W) \approx \mathbb{R}$. Different choices of $V_{ij}$, or isomorphism $W \to V_{ij}$, yield a matrix similar to $M_i$. Each eigenvalue of $M_i$ of multiplicity $r$ gives an eigenvalue of $A_{ii}$, and so of $A$, of multiplicity $r$ degree($\mathfrak{v}_i$).

**Fact 1.** *(Notations and assumptions as above.) If $A$ is the Hessian, all eigenvalues are real and each eigenvalue of $M_i$ of multiplicity $r$ will be an eigenvalue of $A$ with multiplicity $r\ degree(\mathfrak{v}_i)$. In particular, $A$ has most $\sum_{i \in [m]} p_i$ distinct real eigenvalues—regardless of the dimension of the underlying space.*

As a brief summary of the strategy, let $\mathcal{F}$ be $G$-invariant. Given a critical point $W$, we compute the isotropy group $G_W$ of $W$. Since the Hessian of $\mathcal{F}$ at $W$ is a $G$-map, may use the isotypic decomposition of the action of $G$ on $M(k, d)$ to compute the spectrum of the Hessian.

**Proof of Theorem 1** Given the above preparation, we prove the theorem in two main steps, assuming equal-width layers for convenience.

First, we give an explicit realization of the isotypic components $\rho_d^\star$ and bound the number of distinct eigenvalues and their multiplicities for isotropy groups containing subgroups of the form $\{e\} \times S_{d-p}$.

Then, we relate the index of a permutation group to the necessity of it containing a subgroup of the form $\{e\} \times S_{d-p}$, thereby concluding the proof.

The first step can be bypassed by invoking stabilization properties, as in Arjevani (2024), which immediately yield (without bias terms):

$$\mathbb{W} \approx 2V_{(d-p-2,1,1)} \oplus 2V_{(d-p-2,2)} \oplus (4p+7)V_{(d-p-1,1)} \oplus (2p^2+5p+5)V_{(d-p)}. \tag{B.8}$$

Nonetheless, we provide an explicit realization of the isotypic decomposition with representatives chosen following Arjevani (2023) (cf. Arjevani & Field (2021b; 2022)), as it is required for implementing the method for computing the Hessian spectrum using only $O(1)$ entries.

ISOTYPIC DECOMPOSITION OF $(\rho_d^\star, \mathbb{W})$

We begin by presenting the decomposition of $M(d, d)$ with respect to the diagonal action of $S_d$ (rows and columns are permuted simultaneously). Let $\mathbb{D}_d$ denote the space of diagonal $d \times d$ matrices, $\mathbb{A}_d$ the space of skew-symmetric $d \times d$ matrices, and $\mathbb{S}_d$ the space of symmetric $d \times d$ matrices with zero diagonal. We have $M(d, d) = \mathbb{D}_d \oplus \mathbb{S}_d \oplus \mathbb{A}_d$. For $d \geq 4$,

- $\mathbb{D}_d$ is the orthogonal $S_d$-invariant direct sum $\mathbb{D}_{d,1} \oplus \mathbb{D}_{d,2}$, where

  1. $\mathbb{D}_{d,1}$ is the space of diagonal matrices with all entries equal and is naturally isomorphic to $V_{(d)}$.
  2. $\mathbb{D}_{d,2}$ is the $(d-1)$-dimensional space of diagonal matrices with diagonal entries summing to zero and is naturally isomorphic to $V_{(d-1,1)}$.

  In particular, the isotypic decomposition of $(\mathbb{D}_d, S_d)$ is $V_{(d)} + V_{(d-1,1)}$.

- $\mathbb{A}_d$ is the orthogonal $S_d$-invariant direct sum $\mathbb{A}_{d,1} \oplus \mathbb{A}_{d,2}$, where

  1. $\mathbb{A}_{d,1}$ is the $(d-1)$-dimensional space of matrices $[a_{ij}]$ for which there exists $(x_1, \cdots, x_d)$ with entries summing to one such that for all $i, j \in [d]$, $a_{ij} = x_i - x_j$,
  2. $\mathbb{A}_{d,2}$ consists of all skew-symmetric matrices with row sums zero.

  As representations, $(\mathbb{A}_{d,1}, S_d)$ is isomorphic to $V_{(d-1,1)}$ and $(\mathbb{A}_{d,2}, S_d)$ is isomorphic to $V_{(d-2,1,1)}$. In particular, the isotypic decomposition of $(\mathbb{A}_d, S_d)$ is $V_{(d-1,1)} + V_{(d-2,1,1)}$.

- $\mathbb{S}_d$ is the orthogonal $S_d$-invariant direct sum $\mathbb{S}_{1,d} \oplus \mathbb{S}_{2,d} \oplus \mathbb{S}_{3,d}$, where

  1. $\mathbb{S}_{d,1}$ is the 1-dimensional space of symmetric matrices with diagonal entries zero and all off diagonal entries equal.
  2. $\mathbb{S}_{d,2}$ is the $(d-1)$-dimensional space of matrices $[a_{ij}] \in \mathbb{S}_d$ for which there exists $(x_1, \cdots, x_d)$ with entries summing to one such that for all $i, j \in [d]$, $i \neq j$, $a_{ij} = x_i + x_j$.
  3. $\mathbb{S}_{d,3}$ consists of all symmetric matrices in $\mathbb{S}_d$ with all row (equivalently, column) sums zero.
  4. $\dim(\mathbb{S}_{d,3}) = \frac{d(d-3)}{2}$.

  The representations $(\mathbb{S}_{d,i}, S_d)$ are irreducible, $i \in [3]$: $(\mathbb{S}_{d,1}, S_d)$ is isomorphic to the trivial representation $V_{(d)}$, $(\mathbb{S}_{d,2})$ is isomorphic to the standard representation $V_{(d-1,1)}$ and $(\mathbb{S}_{d,3}, S_d)$ is isomorphic to $V_{(d-2,2)}$.

Since $\mathbb{W} = M(d, d) \oplus M(d, d) \oplus \mathbb{R}^d$, the weight space decomposes into irreducible $S_d$-subrepresentations under the diagonal action of $\rho_d^\star$. Explicitly, we have

$$2V_{(d-2,1,1)} + 2V_{(d-2,2)} + 7V_{(d-1,1)} + 5V_{(d)}.$$

Regard $S_{d-p}$ as embedded in $S_d$ via the inclusion $\{e\} \times S_{d-p}$. Next, a $G$-map with $S_{d-p} \subseteq G \subseteq S_d$ is in particular a $S_{d-p}$-map. To analyze the decomposition under the action of subgroups of the form $S_{d-p} \subseteq S_d$ induced from $S_d$, we proceed inductively by splitting factors $p$ times in total. The case of two factors serves as the base step. Assume $p + q = d$, regard $S_p \times S_q$ as a subgroup of $S_d$ and restrict the diagonal action of $S_d$ on $M(d, d)$ to $S_p \times S_q$ to define $M(d, d)$ as an $S_p \times S_q$-space. Clearly, $M(d, d)$ decomposes as an orthogonal $S_p \times S_q$-invariant direct sum

$$M(d, d) = M(p, p) \oplus M(p, q) \oplus M(q, p) \oplus M(q, q),$$

where $M(p, p)$ is an $S_p$-space and $M(q, q)$ is an $S_q$ space (diagonal actions). We regard $M(p, q)$ and $M(q, p)$ as $S_p \times S_q$-spaces. Thus, $S_p$ acts on $M(p, q)$ (resp. $M(q, p)$) by permuting rows (resp. columns) and $S_q$ acts on $M(p, q)$ (resp. $M(q, p)$) by permuting columns (resp. rows). Extending this analysis to the full weight space $\mathbb{W} = M(d, d) \oplus M(d, d) \oplus \mathbb{R}^d$ yields the corresponding decomposition under $S_p \times S_q$, which aligns with the structure stated in (B.8).

**Relating index to structure**    A earlier, given $\Delta \subseteq [d]$, we let $G_{(\Delta)}$ and $G_{\{\Delta\}}$ denote the pointwise and setwise stabilizers of $\Delta$, respectively. By (Dixon & Mortimer, 1996, Theorem 5.2), one of the following holds:

1. for some $\Delta \subset [d]$ with $|\Delta| < p$, $A_{(\Delta)} \subseteq G \subseteq A_{\{\Delta\}}$.

2. $d = 2m$ is even, $G$ is imprimitive with two blocks of size $m$, and $|S_d : G| = \frac{1}{2}\binom{d}{m}$.

3. one of six exceptional cases hold where:

    (a) $G$ is primitive with blocks of size 2 on $[d]$ and $(d, p, |S_d : G|) = (6, 3, 15)$.
    (b) $G$ is primitive on $[d]$ and $(d, p, |S_d : G|) = (6, 2, 6, \mathrm{PGL}_2(5)), (6, 2, 12, \mathrm{PSL}_2(5)), (7, 3, 30, \mathrm{PSL}_3(2))$ or $(8, 3, 30, \mathrm{AGL}_3(2))$.

In Case 1, $S_{d-p} \subseteq G$. By the above, the isotypic decomposition of $M(d - p, d - p)$ is given by $2V_{(d-p)} + 3V_{(d-p-1,1)} + V_{(d-p-2,1,1)} + V_{(d-p-2,2)}$. On $\mathbb{R}^{d-p}$, it holds that $\mathbb{R}^{d-p} = V_{(d-p)} \oplus V_{(d-p-1,1)}$. Consequently, the isotypic decomposition of $\mathbb{W}$ corresponding to $S_{d-p}$ is $2V_{(d-p-1,1,1)} \oplus 2V_{(d-p-2,2)} \oplus (4p + 7)V_{(d-p-1,1)} \oplus (2p^2 + 5p + 5)V_{(d-p)}$, with bias terms adding two copies of $\mathbb{R}^d$ (natural $S_d$-permutation representation). Thus, since Case 2 is ruled out by the constraint on $p$ and Case 3 by the constraint on $d$, the first part of the theorem follows. We remark that although our primary focus is on isotropy groups with $p = \Theta(1)$, the skewed structure of the Hessian spectrum persists in the case of the wreath product, corresponding to $p = d/2$: there exist eigenvalues with multiplicities $\Theta(d^2)$ and $\Theta(1)$. The complete list of irreducible representations of $S_{d/2} \wr S_2$ is provided in James (2006).

As for the second part of the theorem, by restricting the action of $S_{d-p}$ to $M(d - p, d - p) \oplus M(d - p, d - p) \oplus \mathbb{R}^{d-p}$, we observe that under any arbitrarily small (symmetric) perturbation, eigenvalues corresponding to different copies of $S_d$-representation become distinct. This condition is, of course, open and dense, thereby implying genericity; see (Arjevani, 2024, Section 4.1).

A characterization of the shape of the singular spectrum of a flattening of the tensor holding third-order derivatives considered in loc. cit. can be obtained through the formula:

$$\begin{aligned}
\mathrm{Sym}^2((\mathbb{R}^d)^{\otimes 2}) \approx\; & 11V_{(d)} + 21V_{(d-1,1)} + 19V_{(d-2,2)} + 13V_{(d-2,1,1)} + 6V_{(d-3,3)} \\
& + 10V_{(d-3,2,1)} + 4V_{(d-3,1,1,1)} + V_{(d-4,4)} + V_{(d-4,3,1)} + 2V_{(d-4,2,2)} \\
& + V_{(d-4,2,1,1)} + V_{(d-4,1,1,1,1)}.
\end{aligned}$$

$\square$

COEFFICIENT MATRICES BY ISOTYPIC COMPONENT

Below, we present the coefficient matrices used to compute eigenvalues within each isotypic component associated with the isotropy group of the target network, $S_d$. Superscripts indicate the

isomorphism class of the corresponding isotypic component, as determined by an irreducible $S_d$-representation.

$$M_{1,1}^{V(d-2,1,1)} = H_{2,2} - H_{2d+2,2} - H_{3,2} - H_{d+1,2} + 2H_{d+3,2},$$

$$M_{1,2}^{V(d-2,1,1)} = H_{d^2+2,2} + H_{d^2+2d+1,2} - H_{d^2+2d+2,2} - H_{d^2+3,2} - H_{d^2+d+1,2} + H_{d^2+d+3,2},$$

$$M_{2,1}^{V(d-2,1,1)} = H_{d^2+2,2} + H_{d^2+2d+1,2} - H_{d^2+2d+2,2} - H_{d^2+3,2} - H_{d^2+d+1,2} + H_{d^2+d+3,2},$$

$$M_{2,2}^{V(d-2,1,1)} = H_{d^2+2,d^2+2} - H_{d^2+2d+2,d^2+2} - H_{d^2+3,d^2+2} - H_{d^2+d+1,d^2+2} + 2H_{d^2+d+3,d^2+2},$$

$$M_{1,1}^{V(d-2,2)} = H_{2,2} - H_{2d+2,2} + 2H_{2d+4,2} - H_{3,2} + H_{d+1,2} - 2H_{d+3,2},$$

$$M_{1,2}^{V(d-2,2)} = H_{d^2+2,2} - H_{d^2+2d+1,2} - H_{d^2+2d+2,2} + 2H_{d^2+2d+4,2} - H_{d^2+3,2} + H_{d^2+d+1,2} - H_{d^2+d+3,2},$$

$$M_{2,1}^{V(d-2,2)} = H_{d^2+2,2} - H_{d^2+2d+1,2} - H_{d^2+2d+2,2} + 2H_{d^2+2d+4,2} - H_{d^2+3,2} + H_{d^2+d+1,2} - H_{d^2+d+3,2},$$

$$M_{2,2}^{V(d-2,2)} = H_{d^2+2,d^2+2} - H_{d^2+2d+2,d^2+2} + 2H_{d^2+2d+4,d^2+2} - H_{d^2+3,d^2+2} + H_{d^2+d+1,d^2+2} - 2H_{d^2+d+3,d^2+2},$$

$$M_{1,1}^{V(d-1,1)} = H_{1,1} - H_{d+2,1},$$

$$M_{1,2}^{V(d-1,1)} = H_{2,1}d - H_{d+1,1}d,$$

$$M_{1,3}^{V(d-1,1)} = H_{2,1}(d-2) + H_{d+1,1}(d-2) - 2H_{d+3,1}(d-2),$$

$$M_{1,4}^{V(d-1,1)} = H_{d^2+1,1} - H_{d^2+d+2,1},$$

$$M_{1,5}^{V(d-1,1)} = H_{d^2+2,1}d - H_{d^2+d+1,1}d,$$

$$M_{1,6}^{V(d-1,1)} = H_{d^2+2,1}(d-2) + H_{d^2+d+1,1}(d-2) - 2H_{d^2+d+3,1}(d-2),$$

$$M_{2,1}^{V(d-1,1)} = \frac{H_{2,1}}{2} - \frac{H_{d+1,1}}{2},$$

$$M_{2,2}^{V(d-1,1)} = H_{2,2} + \frac{H_{2d+2,2}(d-2)}{2} + \frac{H_{3,2}(d-2)}{2} - H_{d+1,2} - H_{d+3,2}(d-2),$$

$$M_{2,3}^{V(d-1,1)} = -\frac{H_{2d+2,2}(d-2)}{2} + \frac{H_{3,2}(d-2)}{2},$$

$$M_{2,4}^{V(d-1,1)} = \frac{H_{d^2+1,2}}{2} - \frac{H_{d^2+d+2,2}}{2},$$

$$M_{2,5}^{V(d-1,1)} = H_{d^2+2,2} - \frac{H_{d^2+2d+1,2}(d-2)}{2} + \frac{H_{d^2+2d+2,2}(d-2)}{2}$$
$$+ \frac{H_{d^2+3,2}(d-2)}{2} - H_{d^2+d+1,2} - \frac{H_{d^2+d+3,2}(d-2)}{2},$$

$$M_{2,6}^{V(d-1,1)} = \frac{H_{d^2+2d+1,2}(d-2)}{2} - \frac{H_{d^2+2d+2,2}(d-2)}{2} + \frac{H_{d^2+3,2}(d-2)}{2} - \frac{H_{d^2+d+3,2}(d-2)}{2},$$

$$M_{3,1}^{V(d-1,1)} = \frac{H_{2,1}}{2} + \frac{H_{d+1,1}}{2} - H_{d+3,1},$$

$$M_{3,2}^{V(d-1,1)} = -\frac{H_{2d+2,2}d}{2} + \frac{H_{3,2}d}{2},$$

$$M_{3,3}^{V(d-1,1)} = H_{2,2} + \frac{H_{2d+2,2}(d-4)}{2} - 2H_{2d+4,2}(d-3) + \frac{H_{3,2}(d-4)}{2} + H_{d+1,2} + H_{d+3,2}(d-4),$$

$$M_{3,4}^{V(d-1,1)} = \frac{H_{d^2+1,2}}{2} - H_{d^2+2d+3,2} + \frac{H_{d^2+d+2,2}}{2},$$

$$M_{3,5}^{V(d-1,1)} = -\frac{H_{d^2+2d+1,2}d}{2} - \frac{H_{d^2+2d+2,2}d}{2} + \frac{H_{d^2+3,2}d}{2} + \frac{H_{d^2+d+3,2}d}{2},$$

$$M_{3,6}^{V(d-1,1)} = H_{d^2+2,2} + \frac{H_{d^2+2d+1,2}(d-4)}{2} + \frac{H_{d^2+2d+2,2}(d-4)}{2} - 2H_{d^2+2d+4,2}(d-3)$$
$$+ \frac{H_{d^2+3,2}(d-4)}{2} + H_{d^2+d+1,2} + \frac{H_{d^2+d+3,2}(d-4)}{2},$$

$$M_{4,1}^{V_{(d-1,1)}} = H_{d^2+1,1} - H_{d^2+d+2,1},$$

$$M_{4,2}^{V_{(d-1,1)}} = H_{d^2+1,2}d - H_{d^2+d+2,2}d,$$

$$M_{4,3}^{V_{(d-1,1)}} = H_{d^2+1,2}(d-2) - 2H_{d^2+2d+3,2}(d-2) + H_{d^2+d+2,2}(d-2),$$

$$M_{4,4}^{V_{(d-1,1)}} = H_{d^2+1,d^2+1} - H_{d^2+d+2,d^2+1},$$

$$M_{4,5}^{V_{(d-1,1)}} = H_{d^2+2,d^2+1}d - H_{d^2+d+1,d^2+1}d,$$

$$M_{4,6}^{V_{(d-1,1)}} = H_{d^2+2,d^2+1}(d-2) + H_{d^2+d+1,d^2+1}(d-2) - 2H_{d^2+d+3,d^2+1}(d-2),$$

$$M_{5,1}^{V_{(d-1,1)}} = \frac{H_{d^2+2,1}}{2} - \frac{H_{d^2+d+1,1}}{2},$$

$$M_{5,2}^{V_{(d-1,1)}} = H_{d^2+2,2} - \frac{H_{d^2+2d+1,2}(d-2)}{2} + \frac{H_{d^2+2d+2,2}(d-2)}{2} + \frac{H_{d^2+3,2}(d-2)}{2} - H_{d^2+d+1,2} - \frac{H_{d^2+d+3,2}(d-2)}{2},$$

$$M_{5,3}^{V_{(d-1,1)}} = -\frac{H_{d^2+2d+1,2}(d-2)}{2} - \frac{H_{d^2+2d+2,2}(d-2)}{2} + \frac{H_{d^2+3,2}(d-2)}{2} + \frac{H_{d^2+d+3,2}(d-2)}{2},$$

$$M_{5,4}^{V_{(d-1,1)}} = \frac{H_{d^2+2,d^2+1}}{2} - \frac{H_{d^2+d+1,d^2+1}}{2},$$

$$M_{5,5}^{V_{(d-1,1)}} = H_{d^2+2,d^2+2} + \frac{H_{d^2+2d+2,d^2+2}(d-2)}{2} + \frac{H_{d^2+3,d^2+2}(d-2)}{2} - H_{d^2+d+1,d^2+2} - H_{d^2+d+3,d^2+2}(d-2),$$

$$M_{5,6}^{V_{(d-1,1)}} = -\frac{H_{d^2+2d+2,d^2+2}(d-2)}{2} + \frac{H_{d^2+3,d^2+2}(d-2)}{2},$$

$$M_{6,1}^{V_{(d-1,1)}} = \frac{H_{d^2+2,1}}{2} + \frac{H_{d^2+d+1,1}}{2} - H_{d^2+d+3,1},$$

$$M_{6,2}^{V_{(d-1,1)}} = \frac{H_{d^2+2d+1,2}d}{2} - \frac{H_{d^2+2d+2,2}d}{2} + \frac{H_{d^2+3,2}d}{2} - \frac{H_{d^2+d+3,2}d}{2},$$

$$M_{6,3}^{V_{(d-1,1)}} = H_{d^2+2,2} + \frac{H_{d^2+2d+1,2}(d-4)}{2} + \frac{H_{d^2+2d+2,2}(d-4)}{2} - 2H_{d^2+2d+4,2}(d-3)$$
$$+ \frac{H_{d^2+3,2}(d-4)}{2} + H_{d^2+d+1,2} + \frac{H_{d^2+d+3,2}(d-4)}{2},$$

$$M_{6,4}^{V_{(d-1,1)}} = \frac{H_{d^2+2,d^2+1}}{2} + \frac{H_{d^2+d+1,d^2+1}}{2} - H_{d^2+d+3,d^2+1},$$

$$M_{6,5}^{V_{(d-1,1)}} = -\frac{H_{d^2+2d+2,d^2+2}d}{2} + \frac{H_{d^2+3,d^2+2}d}{2},$$

$$M_{6,6}^{V_{(d-1,1)}} = H_{d^2+2,d^2+2} + \frac{H_{d^2+2d+2,d^2+2}(d-4)}{2} - 2H_{d^2+2d+4,d^2+2}(d-3)$$
$$+ \frac{H_{d^2+3,d^2+2}(d-4)}{2} + H_{d^2+d+1,d^2+2} + H_{d^2+d+3,d^2+2}(d-4),$$

$$M_{1,1}^{V_{(d)}} = H_{1,1} + H_{d+2,1}(d-1),$$

$$M_{1,2}^{V_{(d)}} = H_{2,1}(d-1) + H_{d+1,1}(d-1) + H_{d+3,1}(d^2-3d+2),$$

$$M_{1,3}^{V_{(d)}} = H_{d^2+1,1} + H_{d^2+d+2,1}(d-1),$$

$$M_{1,4}^{V_{(d)}} = H_{d^2+2,1}(d-1) + H_{d^2+d+1,1}(d-1) + H_{d^2+d+3,1}(d^2-3d+2),$$

$$M_{2,1}^{V^{(d)}} = H_{2,1} + H_{d+1,1} + H_{d+3,1}(d-2),$$

$$M_{2,2}^{V^{(d)}} = H_{2,2} + H_{2d+2,2}(d-2) + H_{2d+4,2}(d^2 - 5d + 6) + H_{3,2}(d-2) + H_{d+1,2} + 2H_{d+3,2}(d-2),$$

$$M_{2,3}^{V^{(d)}} = H_{d^2+1,2} + H_{d^2+2d+3,2}(d-2) + H_{d^2+d+2,2},$$

$$M_{2,4}^{V^{(d)}} = H_{d^2+2,2} + H_{d^2+2d+1,2}(d-2) + H_{d^2+2d+2,2}(d-2) + H_{d^2+2d+4,2}(d^2 - 5d + 6)$$
$$+ H_{d^2+3,2}(d-2) + H_{d^2+d+1,2} + H_{d^2+d+3,2}(d-2),$$

$$M_{3,1}^{V^{(d)}} = H_{d^2+1,1} + H_{d^2+d+2,1}(d-1),$$

$$M_{3,2}^{V^{(d)}} = H_{d^2+1,2}(d-1) + H_{d^2+2d+3,2}(d^2 - 3d + 2) + H_{d^2+d+2,2}(d-1),$$

$$M_{3,3}^{V^{(d)}} = H_{d^2+1,d^2+1} + H_{d^2+d+2,d^2+1}(d-1),$$

$$M_{3,4}^{V^{(d)}} = H_{d^2+2,d^2+1}(d-1) + H_{d^2+d+1,d^2+1}(d-1) + H_{d^2+d+3,d^2+1}(d^2 - 3d + 2),$$

$$M_{4,1}^{V^{(d)}} = H_{d^2+2,1} + H_{d^2+d+1,1} + H_{d^2+d+3,1}(d-2),$$

$$M_{4,2}^{V^{(d)}} = H_{d^2+2,2} + H_{d^2+2d+1,2}(d-2) + H_{d^2+2d+2,2}(d-2) + H_{d^2+2d+4,2}(d^2 - 5d + 6)$$
$$+ H_{d^2+3,2}(d-2) + H_{d^2+d+1,2} + H_{d^2+d+3,2}(d-2),$$

$$M_{4,3}^{V^{(d)}} = H_{d^2+2,d^2+1} + H_{d^2+d+1,d^2+1} + H_{d^2+d+3,d^2+1}(d-2),$$

$$M_{4,4}^{V^{(d)}} = H_{d^2+2,d^2+2} + H_{d^2+2d+2,d^2+2}(d-2) + H_{d^2+2d+4,d^2+2}(d^2 - 5d + 6)$$
$$+ H_{d^2+3,d^2+2}(d-2) + H_{d^2+d+1,d^2+2} + 2H_{d^2+d+3,d^2+2}(d-2).$$

## C    STATISTICAL GENERALIZATION BOUNDS

The preliminaries, as well as the proof of the theorem, follow the presentation in Bartlett et al. (2019).

**Definition 1.** *Let $\mathcal{H}$ denote a class of functions $\mathcal{X} \to \{0, 1\}$ (the* hypothesis class*):*

- *The growth function of $\mathcal{H}$, $\Pi_{\mathcal{H}} : \mathbb{N} \to \mathbb{N}$ is defined by,*

$$\Pi_{\mathcal{H}}(m) := \max_{x_1 \ldots, x_m \in \mathcal{X}} |\{h(x_1), \ldots, h(x_m)\} \ : \ h \in \mathcal{H}\}|$$

- *The Vapnik-Chervonenkis dimension of $H$ is $\mathrm{VCdim}(H) := \sup\{m \in \mathbb{N} \mid \Pi_H(m) = 2^m\}$.*

- *For a class $\mathcal{F}$ of real-valued functions, we define the* subgraph *class $\mathcal{F}'$ from $\mathcal{X} \times \mathbb{R}$ to $\{0, 1\}$ as $\mathcal{F}' := \{sgn(f(x) - r) \mid f \in \mathcal{F}, r \in \mathbb{R}\}$. The pseudodimension of $\mathcal{F}$ is defined as $\mathrm{VCdim}(\mathcal{F}')$.*

PROOF OF THEOREM C

The proof follows the same line as that of (Anthony & Bartlett, 2009, Theorem 1) and (Bartlett et al., 2019, Theorem 6), with $\mathcal{F}$ denoting the class of networks defined in (2.1). Restricting the analysis to fixed-point subspaces of the weight space implies that the weights associated with certain edges in the computation graph of the network are not trainable, and that some weights are shared across different edges. This necessitates, albeit without introducing substantial complications, a formal adaptation of the original proof and the rates.

We need the following lemma.

**Lemma 3.** *Let $p_1, \ldots, p_m$ be polynomoials of degree at most $d$ in $n \leq m$ variables. Then $\{(sgn(p_1(\mathbf{x})), \cdots, sgn(p_m(\mathbf{x}))) \mid \mathbf{x} \in \mathbb{R}^n\}| \leq 2(2emd/n)^n$.*

If $\mathcal{F}$ is a class of functions defined on $\mathcal{X}$, we write $\mathcal{F}|_{\mathcal{Y}}$ for the restriction of the domain of each function in $\mathcal{F}$ to $\mathcal{Y} \subseteq \mathcal{X}$.

**Proposition 3.** *Let $G \leq S_d$ be an isotropy group for the representation $\rho^*$ corresponding to the action of $\Gamma$, and assume that $\mathbb{W}^G$ is $k$ dimensional. Then, restricted to $\mathbb{W}^G$*

$$\mathrm{Pdim}(\mathcal{F}|_{\mathbb{W}^G}) \leq 4 + (4k + 1) \log_2(28ed \log_2(14ed)).$$

*Proof.* We may equivalently bound the VC dimension of the subgraph class $\mathcal{H} = \mathcal{F}|'_{\mathbb{W}^G}$. We assume that hypotheses in $\mathcal{F}|'_{\mathbb{W}^G}$ are represented by networks $N'$, obtained by augmenting $N$ defined in (2.1) with an output unit that computes $\mathrm{sgn}(N(\mathbf{x}; (W^{(1)}, W^{(2)})) - r)$, where $r \in \mathbb{R}$ is an additional input to $N'$. We further assume that all layers have equal width for simplicity.

Assume $\mathbb{W}^G$ is spanned by $k$ pairs of matrices $E_i \in \mathbb{W}$. Accordingly, the weight matrices are given by $W^{(l)} = \sum_{j=1}^k \xi_j E_j^{(l)}, l \in \{1, 2\}$, and the networks considered take the form $N(\mathbf{x}; \xi_j) := N(\mathbf{x}; (W^{(1)}, W^{(2)}))$. Fix $\mathbf{x}_1, \cdots, \mathbf{x}_m \in \mathbb{R}^d$, and define $q_{i,j}^{(1)}(\mathbf{x}) := W_{j,*}^{(1)} \mathbf{x}_i, i \in [m], j \in [d]$. By Lemma 3, the number of tuples of signs induced by these (linear) polynomials is bounded by $2(2emd/k)^k$. Partition the weight space $\mathbb{W}$ to $N_1 := 2(2emd/k)^k$ semialgebraic sets respecting the signs. Restricted to any set in the partition, $\sigma(W_{j,*}^{(1)} \mathbf{x}_i)$ are linear functions in $\xi_j$, and $q_{i,j,l}^{(2)}(\mathbf{x}) := W_{j,*}^{(2)} \sigma(W^{(1)} \mathbf{x}_i) + \mathbf{x}_i, i \in [m], j \in [d]$ are multivariate polynomials of at most degree 2 in $\xi_j$. By Lemma 3 again, the total number of tuples of signs is bounded by $N_1 N_2$ with $N_2 = 2(4emd/k)^k$, and we may refine the previous partition accordingly. Restricted to any of the $N_1 N_2$ sets in the refined partition, $N(\mathbf{x}_i; \xi_j), i \in [m]$, is a multivariate polynomials of at most degree 2 in $\xi_j$. Hence, applying Lemma 3 once more, the total number of tuples of signs is bounded by $N_1 N_2 N_3$ with $N_3 = 2(4em/k)^k$. Applying the same argument to the last output neuron computing thresholds, and letting $N_4 = 2(4em/(k+1))^{k+1}$, we find that

$$\Pi_{\mathcal{H}}(m) \leq N_1 N_2 N_3 N_4$$

$$= 2(2emd/k)^k \cdot 2(4emd/k)^k \cdot 2(4em/k)^k \cdot 2(4em/(k+1))^{k+1} \quad \leq 16 \left( \frac{6emd + 8em}{4k + 1} \right)^{4k+1},$$

where the last inequality follows from the weighted AM–GM inequality. By definition, we have

$$2^{\mathrm{VCdim}(\mathcal{H})} = \Pi_{\mathcal{H}}(\mathrm{VCdim}(\mathcal{H})) \leq 16 \left( \frac{(6emd + 8em)\mathrm{VCdim}(\mathcal{H})}{4k + 1} \right)^{4k+1}$$

By (Bartlett et al., 2019, Lemma 16),

$$\mathrm{VCdim}(\mathcal{H}) \leq 4 + (4k + 1)\log_2(28ed\log_2(14ed)).$$

$\square$

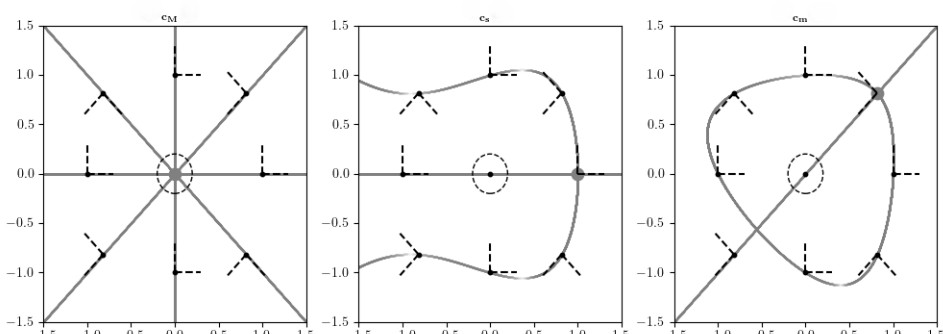

Figure 4: Tangency sets of $h$ defined in (D.9) relative to different critical points $\mathbf{c}_\bullet$. The tangency arcs *break* the symmetry of $\mathbf{c}_\bullet$ to a varying degree. Critical points connected by tangency arcs are at least as symmetric as the arcs and so are themselves symmetry breaking.

# D    TANGENCY SET, JET EQUIVARIANT TRANSVERSALITY AND O-MINIMAL THEORY

## D.1    TANGENCY ARC FOR UNDER HYPEROCTAHEDRAL GROUP

The example below is taken from (Arjevani, 2024, Section 2). To demonstrate the type of results holding under symmetry, consider the following simple $B_2$-invariant function, $B_n$ generally denoting the hyperoctahedral group (Weyl group of type $B$),

$$h(x, y) = x^4 + x^2 y^2 + y^4 - 2x^2 - 2y^2. \tag{D.9}$$

The function $h$ is invariant under $(x, y) \mapsto (y, x)$, $x \mapsto -x$ and $y \mapsto -y$, hence $B_2$-invariant. Computing, we find that $h$ has nine critical points: $\mathbf{c}_M = (0, 0)$ (a maximum), $\mathbf{c}_s = (1, 0)$ giving four critical points (saddles) in total owing to its *orbit* $B_2(1, 0) = \{g(1, 0) \mid g \in B_2\}$, and $\mathbf{c}_m = (\sqrt{3/2}, \sqrt{3/2})$ (a minimum) again yielding $|B_2(\sqrt{3/2}, \sqrt{3/2})| = 4$ critical points. By equivariance, it suffices to compute the tangency set for the orbit representatives: $\mathbf{c}_M, \mathbf{c}_s$ and $\mathbf{c}_m$. Although in this case the tangency set can be described by simple algebraic expressions, our interest lies rather in the geometric situation. Referring to Figure 4, the isotropy groups of tangency arcs are subgroups of $(B_2)_{\mathbf{c}_\bullet}$—arcs having nontrivial isotropy give a lower bound on the isotropy of the critical points to which they connect.

**Symmetry breaking through polynomials**    Polynomials $p$ are a good starting point for building intuition about symmetry breaking, highlighting the role played by gradient equivariance: if $H \subseteq G$ is a subgroup then we may consider the restriction of $\nabla p$ to $V^H$. Briefly, if $p$ has finitely many critical points, then restricting to $V^G$ proves that *generically* certain number of critical points must populate $V^G$ (whether affine real, affine complex or lying at infinity in the associated projective space is a delicate matter, see Arjevani (2024)). Of course, in non-trivial situations $\dim V^G < \dim V$, forcing by the same argument critical points to populate fixed point spaces lying directly below $V^G$ in the lattice of fixed point spaces (associated in a Galois connection with the lattice of isotropy subgroups of $G$). The process is iterated until all critical points have been exhausted.

**Definition 2** (Notation & assumptions as above.)**.** *A critical point of a $G$-invariant function is $m$-SB if the shortest path from its isotropy subgroup to $G$, within the lattice of* isotropy *subgroups, has length $m$.*

Although $S_2 \times S_2 \subseteq D_4 \subseteq S_4$ and $S_3 \times S_3 \subseteq S_3 \wr S_2 \subseteq S_6$ (suitably embedded), for the permutation action of $S_d$, both $S_2 \times S_2$ and $S_3 \times S_3$ are 1-SB, rather than 2-SB, as both intermediate groups are transitive, entailing full symmetry with isotropy group $S_d$. For the diagonal action, however, transitivity does not imply full symmetry, nor does primitivity; rather, 2-transitivity does. Thus, the affine group $\mathrm{AGL}(V)$ never occurs as isotropy group for this action, but $S_3 \wr S_2$, as shown later, both imprimitive and primitive products, do. We find that, contrary to the permutation action, $S_3 \wr S_2$

(imprimitve) is 1-SB for the diagonal action, and $S_3 \times S_3$ 2-SB. Owing to CFGS, we have complete knowledge of all finite 2-transitive groups; all may be readily eliminated from the list of candidate isotropy groups.

**Remark 1.** *The orbits of the action of a subgroup $H \subseteq S_d$ induced on $[d]^2$ are referred to as* orbitals; *their count determines the* rank *of $H$ (i.e., the dimensionality of the associated fixed point space). Consequently, points with isotropy $H$ exhibit full symmetry under the diagonal action, as implied by 2-transitivity, iff $H$ has rank 2. For the discussion above, generalized through the use of rank, see Arjevani (2024).* ✠

**Example 5.** *Define a cubic homogeneous polynomial $p_3(x_1, \ldots, x_{d^2}) = \frac{1}{3} \sum x_i^3 - \frac{1}{2} \sum x_i^2$ over $V = M(d, d)$, with entries enumerated row-wise. Clearly, $p_3$ is invariant under all representations indicated in Example 3, and has $2^d$ critical points, each corresponding to a subset of $\{1, \ldots, d\}$ determining which entries are ones, with all others being zero. The isotropy group of a critical point depends on the representation considered. A natural way to compute the distribution of the isotropy groups is to begin with points that exhibit full symmetry and gradually descend the lattice of subgroups until all critical points have been identified. Below, we give the details for $M(2, 2)$.*

*When regarded as an $(\rho_{S_4}, M(2, 2))$-invariant polynomial, $V^{S_4} = \mathbb{R}1_{2 \times 2}$, has two critical points: $0_{2 \times 2}$ and $1_{2 \times 2}$. All transitive subgroups of $S_4$ may be now disregarded on account of inducing the same fixed point space as $S_4$). This leaves us with the intransitive case, see Section B, which is easily verified to exhaust all reamining critical points. In particular all, but two points, are 1-SB.*

*Regarded as an $(\rho_{S_2 \times S_2}, M(2, 2))$-invariant polynomial, $V^{S_2 \times S_2} = V^{S_4}$, hence holding the same two critical points (careful attention must be given to the 'ambient' representation in each side of the equation as the action of $S_4$ restricted to $S_2 \times S_2$ differs from the action defined by the representation $(\rho_{S_2 \times S_2}, M(2, 2))$.). The remaining critical points are located through the subgroup lattice computed in Example 1. Thus, $V^{\langle e, (12) \rangle}$ is two-dimensional spanned by $\mathbf{e}_1 \mathbf{1}_2^\top$ and $\mathbf{e}_2 \mathbf{1}_2^\top$ giving four critical point, and an analogous situation holds for $V^{\langle (12), e \rangle}$ and $V^{\langle (12), (12) \rangle}$. The three subspaces intersect at $V^{S_2 \times S_2}$—thus the totality of critical points is organized by an inclusion–exclusion principle. Contrary to the previous case, here, 2 points are 0-SB, 6 are 1-SB, and the remaining 8 are 2-SB (trivial symmetry).*

*Regarded yet as an $(\rho_{S_2}, M(2, 2))$-invariant polynomial, $V^{S_2}$ is a two-dimensional space spanned by $I_2$ and $1_{2 \times 2} - I_2$, holding four critical points corresponding to binary combinations of the basis matrices. Observe however that while more points exhibit full symmetry in this case, all other 12 points possess trivial symmetry.*

*As $d$ increases, the variety of subgroup types required to describe the symmetry of critical points grows richer, reflecting, among other aspects, number-theoretic characteristics of the group order. For instance, one finds subgroups that are transitive for the permutation action but are intransitive for the diagonal action of $S_4$, e.g., $I_2 \otimes (1_{2 \times 2} - I_2)$ with isotropy $S_2 \wr S_2 \cong D_4$, the dihedral group of order 8.* ※

| **SB** | $S_3 \times S_3$ | | $S_3$ | |
|--------|------------------|-----|-------|-----|
| **0** | $S_3 \times S_3$ | 2 | $S_3$ | 4 |
| **1** | $D_6$ | 12 | $\mathbb{Z}_2, \mathbb{Z}_3$ | 88 |
| **2** | $\mathbb{Z}_2 \times \mathbb{Z}_2, S_3$ | 102 | $\{e\}$ | 420 |
| **3** | $\mathbb{Z}_2$ | 252 | | |
| **4** | $\{e\}$ | 144 | | |

Table 4: Isotropy types and count of critical points of $p_3$, analyzed in Example 5, by their symmetry breaking level.

Higher-degree homogeneous polynomials have a larger number of critical points. Simple dimensional considerations suggest that new sturctures and isotropy type may emerge.

**Example 6.** *The homogeneous quartic polynomial $p_4(x_1, \ldots, x_{d^2}) = \frac{1}{4} \sum x_i^4 - \frac{1}{2} \sum x_i^2$ over $M(d, d)$, invaraint under all representations indicated in Example 3, possess a critical point with isotropy $S_3 \wr S_2$, primitive product, for $d = 9$.* ※

For polynomials, the preceding discussion gives a clearer appreciation for the nature by which the intricate interplay between isotropy groups and fixed point spaces may lead to critical points exhibiting high symmetry. However, our primary interest lies rather in the broader context where smooth invariants, or at least ones exhibiting sufficient differentiability, are involved, see Arjevani (2024).

