# OpenReview forum: "Geometry & Optimization of Three-Layer Networks: Symmetry Breaking as a Unifying Principle"
_ICLR.cc/2026/Conference — Submitted to ICLR 2026_

### Official Review · Reviewer_kpY7 · 2025-10-31

**Soundness:** 3
**Presentation:** 2
**Contribution:** 3
**Rating:** 6
**Confidence:** 3

**Summary:**

This paper develops Symmetry Breaking (SB) as a unifying theoretical framework to explain geometric and optimization phenomena observed in training fully-connected three-layer ReLU networks—most notably the Hessian / Gauss–Newton bulk–outlier spectrum and the early concentration of gradients in low-dimensional subspaces. The authors build a universal symmetry space and employ permutation-group representation theory to show that at SB critical points the Hessian and GN matrices lie in a fixed low-dimensional subspace and their spectra display explicit eigenvalue multiplicities. They validate theoretical predictions with experiments and illustrate an example to reconstruct the spectrum from a fixed small number of Hessian entries.

**Strengths:**

1. The paper offers a unified theoretical explanation for several previously observed experimental phenomena, such as the "bulk–outlier structure" of Hessian and "GD occurs on tiny space".

2. It presents a clear asymptotic characterization of the spectral distribution of Hessian.

**Weaknesses:**

1. The results rely heavily on strong symmetry assumptions, making it unclear how the predictions extend to architectures or datasets that lack such symmetries.

2. In the theoretical example, the Hessian structure can be characterized by 37 parameters. However, the generalizability of this result remains uncertain, which could be critical for empirical estimation of the Hessian.

**Questions:**

1. The “flat minimum conjecture” discussed in this paper appears to deviate from its conventional definition. Traditionally, the conjecture contrasts sharp and flat global minima, whereas this paper compares global and non-global minima. Could the authors clarify this distinction?

2. Could the authors provide examples or experiments on datasets with approximate symmetries (e.g., images under local translations) to illustrate the robustness of the symmetry-breaking predictions?

3. See Weakness 2.

---

### Official Review · Reviewer_m16j · 2025-11-01

**Soundness:** 3
**Presentation:** 2
**Contribution:** 3
**Rating:** 4
**Confidence:** 2

**Summary:**

This paper proposes symmetry breaking as a unifying mechanism behind several well-documented spectral and optimization phenomena in deep learning, focusing on fully connected three-layer ReLU networks. Building on recent theoretical work, the authors show that critical points reached by SGD typically retain large isotropy groups, which forces the Hessian and Gauss–Newton matrices to lie in a low-dimensional symmetry-restricted subspace. Using group-representation theory, they prove that these matrices exhibit eigenvalue multiplicities of $\Theta(d²)$, $\Theta(d)$, and $\Theta(1)$, offering a structural explanation for the bulk-outlier spectrum and the concentration of gradients in a low-dimensional subspace. Experiments confirm the predicted spectral patterns and symmetry-breaking behavior.

**Strengths:**

1. The paper is fairly dense, but the authors clearly made a significant effort to improve readability by presenting high-level, informal statements in the main text while deferring technical theorems and proofs to the appendix. I appreciate the care the authors have taken.

2. While I have not checked the proof of the main theorem, the experiments (e.g., Figure 1) appear to verify the central claims.

3. Using symmetry breaking to explain the empirically observed bulk–outlier spectrum and the concentration of gradients in low-dimensional subspaces is both novel and significant.

4. The implicit bias component of the paper may be somewhat overstated, since the concentration of gradients within the fixed-point space at early stages of training is demonstrated empirically rather than theoretically. Nonetheless, the idea is interesting and merits further investigation.

**Weaknesses:**

1. Even though the authors have simplified the exposition, the paper remains extremely dense.

2. Figures 1 and 2 may still be difficult to interpret, even with the lengthy captions. For example, Figure 1(a) would benefit from additional explanation; it is unclear why Figure 1(b) uses density while subsequent panels use multiplicity; and in Figure 1(c), the presence of $O(d)$ eigenvalues with multiplicity $\Theta(d)$ is not immediately apparent.

3. The terminology is somewhat confusing: if a critical point has a large isotropy group, would “symmetry-preserving” not be more accurate than “symmetry-breaking”?

4. The authors state that SGD converges to critical points that break the symmetry of the expected loss in a limited manner. Is this purely empirical, or is there a theoretical justification?

5. Lines 275–286 are difficult to parse. Are the authors claiming that if an SB critical point with a large isotropy group exists, then nearby one can find other SB critical points with slightly smaller isotropy groups? If so, what is the implication? Does this suggest that SGD must converge to them?

6. Typo: W^{\Gamma_W} in line 358 and 368.

7. While the theoretical and empirical observations in this paper are interesting, some claims appear somewhat overstated. It is not always clear whether statements such as (1) SGD converges to SB critical points, (2) gradients remain confined to the fixed-point subspace throughout training, and (3) the implicit bias interpretation based on SB, are rigorously proved, heuristic, or purely empirical.

**Questions:**

Please refer to the previous section. I believe this is a very interesting paper, and with additional clarification from the authors, I would be willing to significantly raise my rating.

---

### Official Review · Reviewer_KcVj · 2025-11-01

**Soundness:** 3
**Presentation:** 3
**Contribution:** 3
**Rating:** 6
**Confidence:** 3

**Summary:**

This paper proposes symmetry breaking (SB) as a unifying principle for understanding the geometry and optimization of fully connected three-layer ReLU networks. It shows that many phenomena in deep learning arise naturally from SB. Using group theory, the authors prove that at SB critical points, both curvature matrices lie in the same fixed low-dimensional space and have predictable eigenvalue multiplicities. This allows the eigenspectrum to be recovered from only a small, fixed set of Hessian entries. The analysis further links SB to implicit bias and tighter generalization bounds, suggesting that networks trained by SGD tend to converge to highly symmetric, low-dimensional solutions that generalize better.

**Strengths:**

- The paper presents a coherent and original theoretical framework that links symmetry breaking to the spectral and geometric structure of neural network loss landscapes.
- The mathematical analysis is rigorous and technically impressive, demonstrating a clear command of group representation theory while keeping the results relevant to practical deep learning behavior.
- The empirical results strongly support the theoretical predictions, with measured Hessian and Gauss-Newton spectra matching the multiplicities predicted by the theory.
- The study provides a convincing explanation for why optimization in deep networks occurs in low-dimensional subspaces, grounding a well-known empirical observation in symmetry principles.

**Weaknesses:**

- The experiments are narrow in scope, relying on small, idealized settings that may not reflect real-world architectures or datasets.
- The paper stops short of providing a practical method to identify or exploit symmetry-breaking structures during training, which limits immediate applicability.
- The connection to generalization remains conceptual, with no experiments demonstrating improved generalization or predictive performance.
- The discussion of optimization dynamics is preliminary, and the proposed link between symmetry-breaking subspaces and the edge-of-stability behavior would benefit from deeper empirical validation.

**Questions:**

- Could the authors clarify how to measure or estimate isotropy size in practical training scenarios?
- Could the framework be extended to convolutional or transformer architectures, and what challenges might arise there?
- Is there a systematic way to detect when training has entered a symmetry-breaking regime?

---

### Official Review · Reviewer_4tFM · 2025-11-02

**Soundness:** 2
**Presentation:** 3
**Contribution:** 2
**Rating:** 2
**Confidence:** 3

**Summary:**

This paper looks at learning dynamics through the lens of weight space invariances and argues symmetry breaking plays a key role in it. They show that when gradient descent converges to critical points, these points break the symmetries of the loss function (induced by architecture + data). Using group representation theory, they prove the Hessian and Gauss-Newton matrices at such points must have a characteristic bulk-and-outliers eigenvalue spectrum with specific multiplicities. They argue this explains why gradients concentrate in low-dimensional subspaces and propose symmetry breaking as a form of implicit bias that yields tighter generalization bounds.
I think the idea is quite interesting, but there may be important flaws in the arguments.

**Strengths:**

1. The paper provides a group-theoretic explanation for the bulk-outliers Hessian spectrum using representation theory (Theorem 1). It predicts the correct eigenvalue multiplicities ($\\Theta(d^2), \\Theta(d), \\Theta(1)$) and empirically verifies them.
2. Their symmetry breaking argument makes a nice connection between multiple empirical observations: bulk-outlier spectra, GD being low-dimensional, and the structure of both Hessian and Gauss-Newton matrices. This seems to be a novel idea compared to previous Random Matrix Theory or purely empirical approaches.
3. The paper demonstrates how to compute eigenspectra in extremely high dimensions ($>10^8$ parameters) using only 37 Hessian entries by exploiting symmetry structure (Section 5, Figure 1d). This is a concrete practical contribution beyond the conceptual framework.

**Weaknesses:**

1. **Important Confusion About Slow Modes:** I think a very fundamental point is being missed by focusing on large Hessian eigenvalues: Contrasting the slow modes from symmetries. The paper conflates symmetry-induced degeneracies (which create unimportant near-zero eigenvalues) with optimization-relevant slow modes (small but crucial eigenvalues). The "bulk" eigenvalues they dismiss as unimportant should include the slow-curvature directions that dominate gradient descent's convergence time. From classical optimization, small (non-zero) eigenvalues correspond to flat directions where convergence is slowest. These are the bottleneck for optimization, not irrelevant noise. The paper's claim that "optimization happens in outlier subspaces" (large eigenvalue directions) is backwards: gradient descent naturally handles steep directions well but struggles with flat ones. Their framework seems to dismiss the most optimization-critical modes as "bulk" while elevating the easy-to-optimize modes as "important."
2.  **Contradictions on Flat Minima:** The paper shows global minima have much smaller maximum eigenvalues than local minima (Table 1, Figure 1d-e), supporting the flat minima conjecture. However, this directly contradicts the claim that large eigenvalues are "important" for good optimization—if flat minima generalize better, why would we want optimization dominated by large-eigenvalue directions? This tension between their optimization narrative and generalization results is never resolved.
3. **Limited Scope:** This analysis requires very specific conditions: exact or near-exact symmetries in the joint structure of loss, architecture, and data. The method explicitly breaks down when there are no symmetries (Figure 1f). While appreciate controlled experiments, the restriction to three-layer fully-connected networks with specific synthetic targets (Gaussian inputs, identity-matrix teacher) makes it unclear how much insight transfers to practical deep learning.

**Questions:**

1. Have you thought about using parameter space sysmmetries, $L(gW) = L(W)$, instead of your more restrictive $W=gW$? I think it should be easy to show that these larger symmetries also correspond to zero modes of the Hessian actually, so your argument may apply to them as well.
2. How would you distinguish between true symmetry zero modes and slow modes which result in relatively flat Hessian directions?

---

> ### Author Response · Authors · 2025-11-17
>
> **Important clarification on slow modes.**
>
> Your concerns support the approach presented in this work.
>
> Prior work attempt to explain constrained gradient dynamics via large eigenvalues, whereas we analyze the fixed-point spaces (Figure 3), which include both large and small eigenvalues. You are indicating a potential strength of the work rather than a weakness.
>
> **Flat Minima**\
> Please see above.
>
> Put differently, symmetry of the function yields critical points that lie on low-dimensional fixed-point spaces, through which we argue about:\
> a) a Hessian spectrum with a bulk–outlier structure, \
> b) gradient descent approaching the fixed-point spaces tangentially in certain parameter regimes.
>
> In particular, the paper does not dismiss any eigenvalue as being less or more important than others.
>
> Despite the extensive body of work on these fundamental phenomena, they currently lack an established explanation.\
> Symmetry breaking offers a clean and, in our opinion, compelling one.
>
> **Limited Scope**\
> At present, analyzing modern networks in their full, unconstrained formulation is not within reach. Assumptions such as infinite-width or random-matrix limits serve as reasonable idealizations at this stage of DL theory, much as one may posit nontrivial symmetries in the data.
>
> Q1. $L(gW)=L(W)$ is already the formulation employed in the paper. \
> Q2. The representation-theoretic approach presented in the paper offers one way to address this.
> \
> \
> E pur si muove.

---

### Author Response · Authors · 2025-11-17

We thank the reviewers for their time and feedback.\
Given the very low likelihood of acceptance under the present reviews, we will revise and resubmit.

---

### Meta-Review · Area_Chair_NEgg · 2026-01-05

**Summary:**

While the paper looks interesting, the authors say that they will resubmit, and for this reason, I recommend rejection.

Having quickly skimmed through the paper, I think the authors would benefit greatly from distinguishing their contribution from those in the following two works, which, of course, is only a kind suggestion:
(1) https://arxiv.org/abs/2408.15495
(2) https://arxiv.org/abs/2309.16932

**Reviewer Scores:**

NA

---

### Decision · Program_Chairs · 2026-01-26

Reject